# TAO-AMODAL: A BENCHMARK FOR TRACKING ANY OBJECT AMODALLY

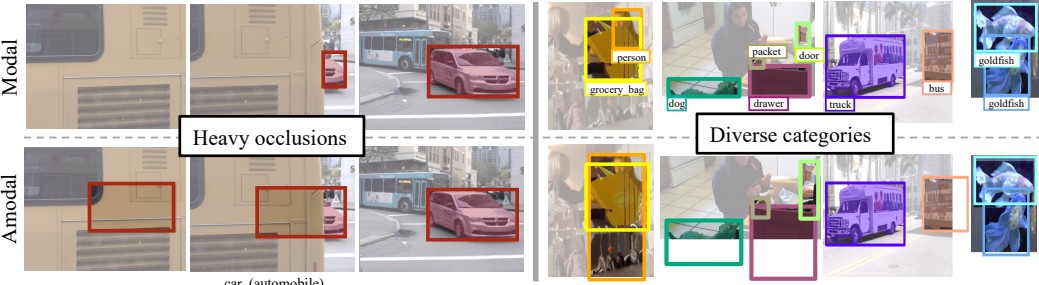

Figure 1: **TAO-Amodal.** We present TAO-Amodal, a dataset of amodal (bounding box) annotations for fully occluded and partially occluded (both within the image frame and out-of-frame) objects in videos from the TAO dataset (Dave et al., 2020a). Our dataset consists of 332k boxes that cover multiple occlusion scenarios across 2,907 videos with annotations for 833 object categories. TAO-Amodal aims at assessing the occlusion reasoning capabilities of current trackers for amodal tracking.

## ABSTRACT

Amodal perception, the ability to comprehend complete object structures from partial visibility, is a fundamental skill, even for infants. Its significance extends to applications like autonomous driving, where a clear understanding of heavily occluded objects is essential. However, modern detection and tracking algorithms often overlook this critical capability, perhaps due to the prevalence of *modal* annotations in most benchmarks. To address the scarcity of amodal benchmarks, we introduce TAO-Amodal, featuring 833 diverse categories in thousands of video sequences. Our dataset includes *amodal* and modal bounding boxes for visible and partially or fully occluded objects, including those that are partially out of the camera frame. We investigate the current lay-of-the-land in both amodal tracking and detection by benchmarking state-of-the-art modal trackers and amodal segmentation methods. We find that existing methods, even when adapted for amodal tracking, struggle to detect and track objects under heavy occlusion. To mitigate this, we explore simple finetuning schemes that can increase the amodal tracking and detection metrics of occluded objects by 2.1% and 3.3%.

## 1 INTRODUCTION

Machine perception, particularly in object detection and tracking, has focused primarily on reasoning about *visible* or modal objects. This modal perception ignores parts of the three-dimensional world that are *occluded* to the camera. However, amodal completion of objects in the real-world (e.g., seeing a setting sun but understanding it is whole) and their persistence over time (e.g., person walking behind a car in Fig. 2) are fundamental capabilities that develop in humans in their early years (Kavsek, 2004; Otsuka et al., 2006; Baillargeon & DeVos, 1991). In autonomous systems, this online amodal reasoning finds a direct application in downstream motion planning and navigation. Despite this, object detection and tracking stacks give little importance to partially or completely

Traditional (modal) detection/tracking

Amodal detection/tracking

Figure 2: **Traditional modal perception (top) *vs.* amodal perception (bottom).** Given a sequence of images, traditional detection and tracking algorithms concentrate on identifying visible segments of multiple objects within the scene. Consequently, they face challenges resulting in perculiar output such as vanishing bounding boxes or tiny box sizes under occlusion scenarios. Amodal perception advances beyond conventional approaches by inferring complete object boundaries, thereby predicting bounding boxes that extend to the full object extent, even when certain portions are occluded.

occluded objects; this becomes apparent in datasets that are only annotated modally (Voigtlaender et al., 2019; Krasin et al., 2017; Gupta et al., 2019; Lin et al., 2014; Everingham et al., 2010; Fan et al., 2019; Yu et al., 2020; Dave et al., 2020a) but are still widely used and built upon by algorithms. These algorithms (Li et al., 2022b; Fischer et al., 2023; Zhou et al., 2022b; Hsieh et al., 2023; Li et al., 2022a; Ren et al., 2015) in turn learn to perceive only modal objects.

To address this gap, we introduce a benchmark for large-scale amodal tracking, which requires estimating the full extent of objects through heavy and even complete occlusions. Our benchmark, TAO-Amodal, annotates 17,000 objects with amodal bounding boxes, along with human confidence estimates, from 833 classes in 2,907 videos. While prior datasets focus on images or are limited to a small vocabulary of classes (Tab. 1), our benchmark evaluates amodal tracking for hundreds of object classes.

We define and address two kinds of occlusions: in-frame and out-of-frame, since objects can get occluded due to other objects in the scene, *and* due to the limited field-of-view of cameras during casual captures. As annotating amodal bounding boxes can be ambiguous and challenging, we design a new annotation protocol with detailed guidelines to improve human annotation. For instance, we ask professional annotators to refer to both preceding and succeeding frames for highly occluded objects. In addition to two rounds of professional check and an additional manual quality check after the annotation process, we ensure the annotations (>99%) maintain a high-level of quality. Importantly, we base our benchmark on a large-vocabulary multi-object tracking dataset, TAO (Dave et al., 2020a). This choice allows us to pair our amodal box annotations with class labels, modal boxes, and precise modal mask annotations (Athar et al., 2023) collected in prior work.

We propose TAO-Amodal primarily as an *evaluation* benchmark with the equipped data and make the 'validation' and 'test' set larger to reliably benchmark trackers (ref. Sec. 3.2). We are not the first to do this: datasets in the multi-object tracking (Dave et al., 2020a) community have similarly focused on evaluation. With the success of foundation models trained on internet data, high quality *evaluation* benchmarks are more important than ever, as evidenced in the NLP community (*e.g.*, MMLU (Hendrycks et al., 2020)). Our training set is constructed in the spirit of instruction-tuning datasets, where only a small amount of data is used to *align* pretrained models to a specific task. Additional analysis on training dataset size (see appendix) validates our choice of dedicating most annotation budget towards robust evaluation.

Table 1: **Statistics of amodal datasets.** TAO-Amodal is proposed as an *evaluation* benchmark for amodal tracking. We compare our dataset to prior image (first block), synthetic video (second block), and real video (last block) datasets. TAO-Amodal is notable for being *real-world* videos that span far *more categories* and far *more annotated frames* for evaluation. Track length is averaged over the dataset in seconds, while total length is the length of eval sequences in seconds. We define heavy occlusion as objects with visibility below 10%, and partial as between 10%-80%. Occluded tracks are those that have heavy or partial occlusions for more than 5 seconds. Out-of-frame (OoF) objects are ones that extend partially beyond the image boundary.

| | | # Sequences | | | | Track | Total | # Occluded Boxes | | | # Occluded | Ann |
| | Total | Test | Val | Train | Classes | length | length | Partial | Heavy | OoF | tracks | fps |
|---|---|---|---|---|---|---|---|---|---|---|---|---|
| COCO-Amodal (Zhu et al., 2017) | 5000 | 1250 | 1250 | 2500 | 5652 | - | - | 34.7k | 1.3k | 0 | - | - |
| Sail-VOS (Hu et al., 2019) | 201 | 0 | 41 | 160 | 162 | 14.14 | 3,359 | 559.5k | 704.8k | 0 | 7.9k | 8 |
| Sail-VOS-3D (Hu et al., 2021b) | 202 | 0 | 41 | 161 | 24 | 13.10 | 2,808 | 295.0k | 387.5k | 0 | 5.0k | 8 |
| NuScenes (Caesar et al., 2019) | 1000 | 150 | 150 | **700** | 23 | 9.06 | 6,000 | 571.1k | **139.5k** | **219k** | **24.5k** | 20 |
| MOT17 (Milan et al., 2016) | 14 | 7 | 0 | 7 | 1 | 6.98 | 248 | 51.2k | 16.4k | 16k | 0.1k | **30** |
| MOT20 (Dendorfer et al., 2020) | 8 | 4 | 0 | 4 | 1 | 20.55 | 178 | **729.4k** | 88.1k | 88k | 1.6k | 25 |
| **TAO-Amodal** | **2907** | **1419** | **988** | 500 | **833** | **22.24** | **88,605** | 158.2k | 35.1k | 139k | 9.6k | 1 |

Given this benchmark, we set out to evaluate the difficulty of amodal tracking using standard metrics, including detection and tracking AP, and variants (Khurana et al., 2021) that evaluate tracking specifically under partial and complete occlusions. As expected, we find that standard trackers trained with modal annotations do not suffice for amodal tracking.

To adapt existing modal trackers into amodal ones, we finetune them on TAO-Amodal. The closest line of work to amodal tracking is amodal segmentation (Zhan et al., 2023; Li & Malik, 2016; Qi et al., 2019). We benchmark recent amodal segmentation algorithms by running a Kalman-Filter based association during post-processing on their predictions. While this addresses the gap between modal and amodal tracking to some extent, the performance is far from good due to the challenging occlusion scenarios in TAO-Amodal. To mitigate this, we explore different but simple finetuning and data-augmentation strategies inspired by prior work (Li & Malik, 2016; Zhu et al., 2017). This lets us set a new baseline on the tasks of amodal detection and tracking.

In summary, our contributions are as follows: (1) we annotate a large-scale dataset of amodal tracks for diverse objects, consisting of 17k objects spanning 833 categories, (2) we adapt evaluation metrics to handle amodal settings, and evaluate state-of-the-art trackers for our new task, and finally, (3) we investigate multiple finetuning and data-augmentation schemes as simple extensions to improve the existing modal tracking algorithms.

## 2 RELATED WORK

Amodal perception has been studied in the past by benchmarks and algorithms, in both the single-frame (detection) and multi-frame (detection and tracking) settings. Since amodal object annotations are hard to obtain due to the uncertainty in human annotations (c.f. prior work (Khurana et al., 2021) on a human vision experiment), the community has depended heavily on synthetic datasets, or real-world datasets with few classes and limited diversity. We provide an overview of this prior wrok in the rest of this section.

### 2.1 BENCHMARKS

**Real-world datasets.** Amodal object annotations for real-world scenes are largely limited to the surveillance and self-driving domains.MOT 15-20 (Leal-Taixé et al., 2015; Milan et al., 2016; Dendorfer et al., 2020) evaluate multi-object tracking on amodal person detections obtained from detectors trained on MOT annotations. However, these amodal annotations are automatically propagated via linear interpolation of annotations in frames where objects are visible. Additionally, the metrics used by MOT weigh all modal and amodal annotations equally. This largely ignores tracking performance on amodal objects, which form only a small fraction of all annotations.

A number of multimodal (images and 3D LiDAR) datasets for autonomous driving have recently become popular. These include ArgoVerse (1.0 and 2.0) (Chang et al., 2019; Wilson et al., 2021),

Waymo (Sun et al., 2020), nuScenes (Caesar et al., 2019) and KITTI (Geiger et al., 2012). These datasets aim to focus on *3D* tasks, and therefore use human annotators to label all objects in 3D to their full extent. In this setting, amodal annotations arise naturally due to the 3D nature of the data. These 3D boxes, when projected onto 2D images, would be useful for amodal perception; unfortunately, these annotations cover only a small number of object classes. Another way to obtain amodal object annotations is in a multi-view setting. Datasets like CarFusion (Reddy et al., 2018) and MMPTrack (Han et al., 2023) follow this data curation scheme, but, due to the cumbersome data collection process, they are limited to only a single or few categories.

In the single-frame setting, COCO-Amodal, Amodal KINS and NuImages (Caesar et al., 2019; Zhu et al., 2017; Qi et al., 2019) contain amodal annotations, but only cover the cases of partial occlusion: complete occlusions can only be recovered with temporal information, which is missing in image datasets. Moreover, the single-frame setting makes it difficult to evaluate the dynamic aspects of amodal tracking due to the absence of temporal context.

**Synthetic datasets.** An alternative approach to the above is use synthetic data generation pipelines to get amodal annotations. SAIL-VOS and SAILVOS-3D (Hu et al., 2019; 2021b) are such datasets that exploit synthetic dataset curation and come with a number of different types of annotations (bounding boxes, object masks, object categories, their long-range tracks, and 3D meshes). Some of these even suit our case of detecting 'out-of-frame' occlusions, where one could project 3D meshes onto the image plane. While the number of categories are slightly larger for these datasets (including others like ParallelDomain (Tokmakov et al., 2021) and DYCE (Ehsani et al., 2018)), the sim-to-real transfer remains a challenge even for modal perception (Chen et al., 2018; Khodabandeh et al., 2019).

## 2.2 ALGORITHMS

**Amodal perception.** Based off of some amodal datasets, there has been a growing interesting in developing algorithms suitable for amodal perception. Some methods aim to track objects with object permanence (Khurana et al., 2021; Tokmakov et al., 2021; 2022; Van Hoorick et al., 2023; Reddy et al., 2022). Previous work also segment objects amodally (Li & Malik, 2016; Zhan et al., 2023; 2020; Follmann et al., 2019; Xu et al., 2023; Ozguroglu et al., 2024). Some approaches utilize prior-frame information (Zhou et al., 2020; Cai et al., 2022; Wu et al., 2021; Stearns et al., 2022; Du et al., 2023; Yang et al., 2024; Gao & Wang, 2023; Wojke et al., 2017; Zhou et al., 2022b). For instance, GTR (Zhou et al., 2022b) employs a transformer-based architecture and uses trajectory queries to group bounding boxes into trajectories. We lean on similar approaches in this work, and devise a mechanism to generate occlusion cases in the flavor of the data augmentation used by GTR, and show that this is essential to the goal of enabling amodal perception.

**Synthetic data augmentation.** Pasting object segments onto images is a commonly used data augmentation technique which has been proven effective in both modal and amodal perception literature. For instance, a line of amodal segmentation literature (Li & Malik, 2016; Ozguroglu et al., 2024; Zhu et al., 2017) creates synthetic amodal data through pasting object segments onto images for training amodal mask prediction heads. We also observe similar strategies in modal perception. Ghiasi et al. (2021) uses simple copy-paste strategy to improve the instance segmentation. Yun et al. (2019) replaces regions of an image with patches from another image and combines their labels. These techniques can generate data on-the-fly without requiring additional labels. Even though the generated data is far from the natural distribution of real-world images, all aforementioned methods are successful fundamentally because of this data augmentation. Inspired by these works, we develop a data augmentation pipeline, paste-and-occlude (PnO), to randomly simulate occlusion scenarios during training for amodal tracking in Sec. 4.3.

## 3 DATASET ANNOTATION AND DESIGN

**Base dataset.** Existing datasets for modal perception are limited either in terms of their diversity, or the vocabulary of classes. To this end, we build upon the modally annotated TAO dataset. It contains bounding box track annotations of 833 object categories at 1FPS spanning a total of 2,921 videos from 7 different data sources (AVA (Gu et al., 2018), Argoverse (Chang et al., 2019), Cha-

rades (Sigurdsson et al., 2016), HACS (Zhao et al., 2019), LaSOT (Fan et al., 2019), BDD100K (Yu et al., 2020), YFCC100M (Thomee et al., 2016)). Bootstrapping from this dataset allows us to add amodal box annotations to an already existing set of multimodal annotations in TAO – i.e., object classes, modal bounding boxes and modal segmentation masks. TAO follows the single-frame detection datasets, such as LVIS and OpenImages (Gupta et al., 2019; Krasin et al., 2017), in adopting a federated annotation protocol for object tracking: i.e., not every object class is exhaustively annotated in every video. These datasets, similar to ours, often feature a large vocabulary of object classes, making exhaustive annotations unfeasible. We refer the reader to (Dave et al., 2020a; Gupta et al., 2019) for details on federated annotation and evaluation setup, and focus here on our amodal annotation of objects in TAO.

**Scope.**   Since annotators can exhibit a large variation in annotating the precise shape of objects while they undergo partial or even complete occlusion, we annotate using bounding boxes instead of segmentation masks to mark the full extent of objects in the visible scene. We define 'in-frame' occlusions as those occurring from the presence of occluders (which may be other dynamic objects, or static scene elements), and 'out-of-frame' occlusions as those resulting from objects leaving the camera field-of-view. We do not label the extent of occlusion in cases where an object may be partially present *behind* the camera (e.g., a person holding the camera who has their hands visible in the image). For labelling 'out-of-frame' occlusions, we need to fix bounds for annotation on the image plane. We ask annotators to work within an *annotation workspace* that extends to twice the image dimensions in consideration, with the image itself horizontally and vertically center-aligned in this workspace. We select the factor of two to ensure that the workspace covers most of the amodal boxes (99.16%) without touching the border. The workspace could be considered as a larger image with padding and is maintained even when data augmentation is applied.

**Annotation Protocol.**   Since object tracks in TAO are modal in nature, extending boxes to account for in-frame and out-of-frame occlusions requires (1) (in the case of partial occlusion) complementing TAO bounding boxes with amodal boxes, and (2) (in the case of complete occlusion) adding new boxes to object tracks for occluded frames. Out of a total of 358,862 boxes in TAO, our annotators modify 266,902 (74.4%) to account for partial occlusions. Further, TAO-Amodal introduces an additional 23,449 bounding boxes for frames where objects were invisible and unlabeled in TAO. These annotations follow the guidelines detailed in the appendix, covering a wide range of both in-frame and out-of-frame occlusion scenarios. Importantly, we only consider occlusion cases where an object has appeared in the scene before. We exclude occlusions where an object might be partially behind the camera or outside the annotation workspace defined above. We require the annotators to refer to both preceding and subsequent frames for occluded objects. Within the strict purview of the guidelines, when an object's location still cannot be discerned confidently by the annotators, annotators are instructed to mark an is_uncertain flag. From the 23,449 boxes for invisible objects, 20,218 (85.8%) boxes are annotated confidently (i.e., without the uncertain flag), indicating that there is inherent uncertainty in localizing objects when they undergo heavy occlusions (similar to prior work (Khurana et al., 2021) which indicates uncertainty in object location under occlusion). Please note that the annotations still allow for reliable benchmarking of amodal trackers as these uncertain objects represent only a marginal fraction ($< 1\%$) of the data. We provide examples of uncertain objects in the appendix.

Finally, equipped with both modal and amodal annotations for all objects, we add a visibility field to the TAO-Amodal annotations, using the overlap (intersection-over-union) between the modal and amodal boxes as a proxy.

**Quality Control.**   We conduct two rounds of professional quality checks on TAO-Amodal annotations: all bounding box annotations are refined twice by annotators. Finally, the authors of this work conducted a manual quality check reviewing 349 tracks from 7 randomly sampled videos, and found only 2 ($<1\%$) tracks without an uncertainty flag to be erroneous. Both tracks were for objects with complete occlusions (visibility 0.0%) in the video. Our analysis show that nearly all inspected tracks ($> 99\%$) are accurate, indicating the high-quality of amodal tracking annotations in TAO-amodal.

### 3.1 DATASET STATISTICS

We compare the statistics of TAO-Amodal to other amodal benchmarks in Tab. 1. For NuScenes, which only categorizes object visibilities into four buckets, we use interpolation to estimate the number of boxes below visibility 0.1 and 0.8. A few amodal datasets are omitted from the table, either because they have been incorporated into TAO-Amodal (Chang et al., 2019; Yu et al., 2020) or because these datasets lack quantified visibilities for categorizing different occlusion scenarios (Cioppa et al., 2022; Sun et al., 2022). TAO-Amodal covers annotations across an extensive 833 categories, which can be used to learn and evaluate object priors in a large-vocabulary setting. Furthermore, TAO-Amodal features a *10×* longer evaluation duration, ensuring a comprehensive evaluation. We also provide class and occlusion distribution in Figs. 9 and 10 in the appendix.

### 3.2 DATASET SPLITS DESIGN FOR EVALUATION BENCHMARK

Following TAO (Dave et al., 2020a), we propose TAO-Amodal primarily as an *evaluation* benchmark using a larger 'validation' and 'test' set. We construct the training set to align modal trackers with amodal data and propose the validation set for empirical analysis. We reserve the testing set for challenge evaluation following TAO (Dave et al., 2020a). Several key factors informed this design decision, which we discuss in detail.

Following the development of foundation models trained on internet data, the emphasis on high-quality *evaluation* benchmarks is increasingly crucial. TAO-Amodal aligns with the concept of "visual instruction tuning" from NLP (Liu et al., 2024; Wei et al., 2021), where a smaller training set is used to align pre-trained models (*e.g.*, modal trackers) with task-specific foundation models (*e.g.*, amodal trackers). Similar advances are seen in finetuning techniques in vision where limited task-specific data is available (Hu et al., 2021a; Zhang et al., 2023).

High-quality benchmarks drive the need for innovation in curating large-scale training data. This mirrors the evolution in large language models, where the introduction of challenging benchmarks (Zhang et al., 2024a; Lu et al., 2023), led to the collection of large-scale training data (Zhang et al., 2024b). Some benchmarks (Hendrycks et al., 2020) do not have a training set and rely solely on the use of "internet as a training set".

In vision, this paradigm shift appears in the use of synthetic data. For instance, amodal segmentation methods (Ozguroglu et al., 2024; Li & Malik, 2016) create synthetic amodal data by pasting object segments onto images. Prominent modal tracking methods (Zhou et al., 2022b; 2020) generate synthetic training videos by random cropping and resizing of static image datasets (Gupta et al., 2019) to compensate for the lack of large vocabulary tracking data. State-of-the-art methods in 3D vision (*e.g.*, monocular depth estimation (Ke et al., 2024) & scene flow (Xiao et al., 2024)) use out-of-distribution synthetic training samples, and beat prior work on real-world evaluation.

Lastly, we note that despite the same small training set of the original TAO dataset, the modal tracking performance has increased from 10.2 to 27.5 Track-AP over the years (Dave et al., 2020b), which includes performance increase on object categories where little to no training data existed. Therefore, we believe that a robust evaluation benchmark can drive the innovation of more powerful architectures and training objectives.

## 4 AMODAL TRACKING

### 4.1 TRADITIONAL AND AMODAL TRACKING

Given a sequence of images $I^1, I^2, ..., I^t$, tracking approaches aim to output modal bounding boxes $b$, trajectory identifiers $\tau$, and class labels $s$ for objects across all frames. If an object is partially occluded, the box marks only the visible extent of the object, as illustrated in Fig. 2. We focus here on amodal trackers, which similarly take as input a sequence of images, but, in addition to the modal tracker outputs, they generate amodal boxes $b_a$, which cover the full extent of occluded objects.

In practice, training an amodal tracker end-to-end is infeasible due to the limited amount of amodal training data. We focus instead on transforming a conventional tracker into an amodal one by leveraging its understanding of modal objects.

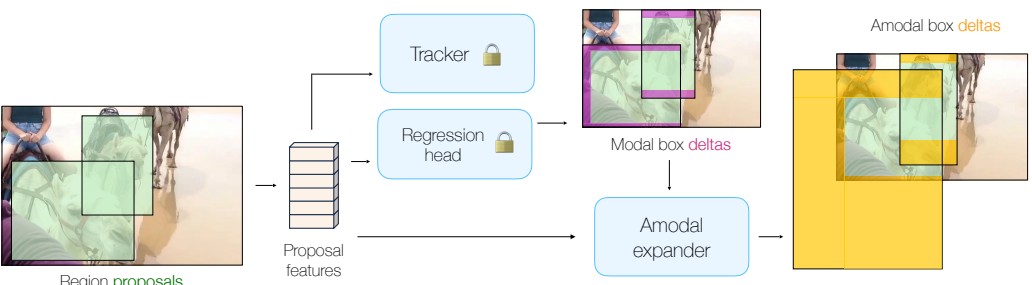

Figure 3: **ROI Head (Girshick, 2015) with Amodal Expander.** Amodal Expander serves as a plug-in fine-tuning scheme to "amodalize" existing detectors or trackers with limited (amodal) training data. It operates by taking as input region proposal features and modal box predictions (often represented as a residual delta with respect to the region proposal) and generates amodal box outputs (again represented as residual deltas). We freeze all modules except the expander during fine-tuning.

## 4.2 AMODAL EXPANDER

We design an amodal expander $E$, which serves as a plug-in module to conventional trackers. For each object, the amodal expander takes as input the modal box $b$ and an embedding $f$ (which can be extracted from the tracker), and generates amodal bounding boxes $b_a$.

**Predicting amodal boxes in a residual manner.** Amodal expander operates as a refinement step, similar to the second stage of two-stage detectors (He et al., 2017) and trackers (Zhou et al., 2022b), and can be applied to most standard modal trackers. We introduce amodal expander on top of GTR (Zhou et al., 2022b). As shown in Fig. 3, GTR produces modal boxes $b$ with corresponding object features $f$, and subsequently refines $b$ through a regression head $R$ by predicting a modal box delta $\Delta b$. Our amodal expander takes as input the modal box delta $\Delta b$ and object feature $f$ as input, generating an amodal box delta. This delta is then applied to the modal proposal $b$ to generate amodal boxes $b_a$, denoted as $E(\Delta b, f) + b \approx b_a$. The training of the amodal expander follows the training of regression head (Ren et al., 2015) by matching box proposals with a ground truth and applying regression loss. We first match modal box predictions $b$ to a **modal** ground truth $b^*$. We then apply the regression loss, smooth L1 (Girshick, 2015), with the corresponding amodal ground truth $b_a^*$:

$$L(b, \Delta b, f) = L_{reg}(E(\Delta b, f) + b, b_a^*) \tag{1}$$

We provide implementation details of amodal expander in the appendix.

## 4.3 SYNTHESIZING OCCLUSION WITH PASTE-AND-OCCLUDE (PNO)

As discussed in Sec. 2.2, object pasting has been shown to be an effective augmentation pipeline in amodal segmentation literature (Ozguroglu et al., 2024) to simulate occlusion scenarios during training. In this work, we develop a similar data augmentation pipeline for amodal tracking, which we refer to as Paste-and-Occlude (PnO). PnO functions by pasting object segments onto the videos to act as occluders. The segment collection comprises 505k objects extracted from LVIS (Gupta et al., 2019) and COCO (Lin et al., 2014) images using segmentation masks. We apply heuristic filtering approach to select these segments to ensure the occluders are not occluded. For each input video, we randomly select 1 to 7 segments from the collection and paste them at arbitrary locations in the starting and last frames, allowing for partial extension beyond the image boundary to replicate out-of-frame situations. The size and location of the segments in the intermediate frames are determined through linear interpolation. Subsequently, we incorporate the ground truth boxes of the pasted segments into the original set of ground truth boxes. We find that PnO leads to improvements in detection across all occlusion scenarios, shown in Sec. 5.3. We posit that this synthetic strategy is particularly important for the long-tailed nature of TAO-amodal, unlike COCO-amodal, where a similar synthetic occlusion strategy leads to limited improvement (Zhu et al., 2017). We provide visual examples of synthetic occlusions and further implementation details in the appendix.

## 5 EMPIRICAL ANALYSIS

In Sec. 5.2, we assess the challenges of amodal detection and tracking by evaluating a number of amodal trackers and segmentors. Next, we investigate fine-tuning strategies and extensions for amodal baselines in Sec. 5.3. We present implementation details, further scaling analysis, and other ablations in the appendix.

### 5.1 EVALUATION METRICS

Using the estimated visibility attributes, we assess the tracking and detection capabilities of the model through variations of detection AP (Lin et al., 2014) and Track-AP (Dave et al., 2020a), representing the average precision across all categories at an IoU threshold of 0.5. We label objects with visibility less than 0.1 as heavily occluded, evaluated as $AP^{[0.0, 0.1]}$, where the superscript indicates the range of object visibility. If the visibility falls between 0.1 and 0.8, we categorize them as partially occluded, while those with visibility greater than 0.8 are considered non-occluded. Objects that extend beyond the image boundary are referred to as out-of-frame (OoF) and evaluated with $AP^{OoF}$. Additionally, we assess the model's performance on modal annotations with Modal AP. In tracking, we evaluate highly or partially occluded tracks (Track-$AP^{[0, 0.8]}$), which are track with visibility at or below 0.8 for more than 5 frames (seconds). We also evaluate performance on modal annotations (Modal Track AP).

The adaptation of AP metrics enables us to align existing trackers for amodal tracking simply using data from TAO-Amodal, as these metrics do not require the model to generate a distribution of bounding boxes as required in *probabilistic* metrics (Khurana et al., 2021). We elaborate on the benefits and limitations of this design choice in Sec. D in the appendix. We follow the federated evaluation setup established in TAO (Dave et al., 2020a) and LVIS (Gupta et al., 2019). Specifically, object classes that are not exhaustively annotated will not be used for computing false positives. We refer the reader to TAO (Dave et al., 2020a) for further details. We summarize the metric definitions in Tab. 5 in the appendix for quick reference.

### 5.2 BENCHMARKING STATE-OF-THE-ART TRACKERS

**Evaluation of modal detectors and trackers.** We use three recent modal trackers, QDTrack (Fischer et al., 2023), TET (Li et al., 2022a) and GTR (Zhou et al., 2022b) and a detector, ViTDet (Li et al., 2022b) for benchmarking. Every modal tracker is pre-trained on either TAO (Dave et al., 2020a) or LVIS (Gupta et al., 2019), ensuring alignment of category vocabulary with our dataset. GTR is trained on the combination of LVIS and COCO (Lin et al., 2014) by generating synthetic videos from static images (Zhou et al., 2020). QDTrack and TET follow similar training procedures, pretraining detectors on LVIS and instance similarity heads on TAO for association. ViTDet is trained on LVIS and combined with online SORT (Bewley et al., 2016) tracker. We also evaluated ViTDet with ByteTrack (Zhang et al., 2022) and observed suboptimal results, which we attribute to its strategy of removing tracks that are not matched in the second frame after their initial appearance. We adapt all models by fine-tuning the regression head on TAO-Amodal training set for 20k iterations and evaluated each model on the validation set. Following TAO (Dave et al., 2020a), we reserve the test set for challenge evaluation.

**Evaluation of off-the-shelf amodal segmentors.** While we finetune modal algorithms for the amodal task, we note that these may not be architecturally optimized for amodal perception. To this end, we evaluate methods from the amodal segmentation line of work (note that mask prediction is more prevalent for amodal perception than box prediction). We benchmark ORCNN (Follmann et al., 2019), Amodal Mask-RCNN (Follmann et al., 2019), AISFormer (Tran et al., 2022) and PCNet (Zhan et al., 2020). ORCNN proposes a loss which brings occluders and ocludees spatially close. Amodal Mask-RCNN trains an additional amodal mask head on top of Mask-RCNN. AISFormer, also based on Mask-RCNN, uses transformer blocks to learn the spatial relations between visible and occluded objects. These methods only need an image as input and are trained on COCOA-cls (Follmann et al., 2019). PCNet takes in modal masks of all objects in the scene as input, and recovers their relative ordering in the scene, before expanding modal masks into amodal ones. We use Detic (Zhou et al., 2022a) to get these modal masks. Finally, we run SORT (Bewley et al.,

Table 2: **Amodal trackers on TAO-Amodal validation set.** We define metrics in Sec. 5.2. The visibility range is indicated by the superscript to denote various levels of occlusion. We fine-tuned modal trackers on TAO-Amodal-train for 20k iterations. Detector (Li et al., 2022b) and amodal segmentation methods (Zhan et al., 2020; Follmann et al., 2019; Tran et al., 2022) were evaluated using Kalman filter based association (Bewley et al., 2016). We evaluated models that predict COCO vocabulary (Lin et al., 2014) using objects within COCO category. GTR is used as the basis for subsequent experiments, considering its performance in detection and tracking metrics. We run all trackers at 1 fps and average AP across categories with an IoU threshold of 0.5.

| Method | FT | Detection Metrics | | | | | Tracking Metrics | |
|---|---|---|---|---|---|---|---|---|
| | | $AP^{[0,0.1]}$ | $AP^{[0.1,0.8]}$ | $AP^{[0.8,1]}$ | $AP^{OoF}$ | AP | AP | $AP^{[0,0.8]}$ |
| PCNet (Zhan et al., 2020) | | 0.48 | 7.15 | 21.43 | 8.69 | 15.59 | 5.80 | 3.91 |
| QDTrack (Fischer et al., 2023) | ✓ | 0.35 | 8.03 | 21.82 | 8.05 | 15.62 | 7.84 | 4.03 |
| TET (Li et al., 2022a) | ✓ | 0.24 | 5.77 | 14.98 | 4.87 | 10.86 | 4.84 | 3.44 |
| ViTDet-B (Li et al., 2022b) | ✓ | 0.77 | 12.57 | 34.33 | 14.18 | 25.94 | 7.66 | 4.38 |
| ViTDet-L (Li et al., 2022b) | ✓ | **1.25** | 15.06 | 38.16 | 15.84 | 29.04 | 9.70 | 5.90 |
| ViTDet-H (Li et al., 2022b) | ✓ | 1.13 | **15.80** | **40.09** | **16.97** | **30.20** | 9.72 | 5.63 |
| GTR (Zhou et al., 2022b) | ✓ | 0.77 | 14.62 | 38.17 | 15.31 | 29.24 | **16.07** | **9.28** |
| COCO category eval | | | | | | | | |
| ORCNN (Follmann et al., 2019) | | 0.33 | 11.78 | 37.88 | 16.68 | 26.09 | 5.72 | 3.43 |
| AmodalMRCNN (Follmann et al., 2019) | | 0.46 | 14.74 | 42.65 | 18.35 | 29.58 | 7.57 | 4.47 |
| AISFormer (Tran et al., 2022) | | 0.36 | 14.23 | 39.76 | 18.61 | 27.70 | 7.88 | 5.90 |
| PCNet (Zhan et al., 2020) | | **1.30** | **20.61** | **53.13** | **24.21** | **37.04** | **11.19** | **8.78** |

Table 3: **Exploring fine-tuning strategies on TAO-Amodal validation set.** We ablate different strategies for finetuning GTR, where the default of tuning regression-head corresponds to the baseline listed in Tab. 2. Finetuning expander modestly outperforms finetuning all or part of the model. Combined with data augmentation, PasteNOcclude (PnO), expander produces noticeable gains for partially occluded and out-of-frame objects. All models (other than the baseline) were trained on TAO-Amodal training set for 20k iterations, while † denotes 45k iterations of training.

| Method | Detection Metrics | | | | | Tracking Metrics | |
|---|---|---|---|---|---|---|---|
| | $AP^{[0,0.1]}$ | $AP^{[0.1,0.8]}$ | $AP^{[0.8,1]}$ | $AP^{OoF}$ | AP | AP | $AP^{[0,0.8]}$ |
| Baseline (GTR (Zhou et al., 2022b)) | 0.78 | 13.24 | 37.54 | 14.18 | 28.19 | 16.02 | 8.86 |
| Fine-tune entire model | 0.52 | 10.36 | 24.08 | 10.34 | 17.93 | 7.70 | 3.93 |
| FT entire model + PnO | 0.79 | 9.68 | 26.56 | 10.10 | 20.16 | 9.05 | 4.30 |
| Fine-tune regression head & proposal network | 0.79 | 10.57 | 27.91 | 11.37 | 21.42 | 9.04 | 4.53 |
| Fine-tune regression head | 0.77 | 14.62 | 38.17 | 15.31 | 29.24 | 16.07 | 9.28 |
| FT regression + PnO | 0.87 | 14.36 | 38.18 | 15.47 | 29.04 | 15.95 | 9.23 |
| Amodal Expander | 0.67 | 16.29 | 37.11 | 17.39 | 29.50 | 16.10 | **10.44** (+1.58) |
| Amodal Expander + PnO | **0.80** (+0.02) | 16.41 | 37.74 | 17.64 | 29.87 | **16.35** (+0.33) | 10.13 |
| Amodal Expander + PnO† | 0.77 | **16.53** (+3.29) | **37.80** (+0.26) | **17.65** (+3.47) | **29.96** (+1.77) | **16.35** (+0.33) | 10.28 (+1.42) |

2016) on top of all boxes obtained from these methods and evaluate them only on COCO classes. PCNet shines likely because it only needs to *expand* the given modal masks.

**How well do SOTA methods handle amodal perception?** In Tab. 2, we see that both amodal segmentation baselines and fine-tuned modal trackers struggle in handling heavy occlusion and out-of-frame cases. To bridge the gap, we further explore different fine-tuning schemes and effects of data augmentation in Tab. 3, introduced in the next section. We report the performance of modal trackers on TAO-Amodal validation set as an ablation in the appendix.

## 5.3 BUILDING AMODAL BASELINES WITH AMODAL EXPANDER

We illustrate amodal expander architecture in Fig. 4 in the appendix. We build the expander on top of GTR (Zhou et al., 2022b) as this method shows reasonable performance in both detection and tracking aspects in Tab. 2, likely due to its transformer-based association architecture that links identities over longer time periods with a sliding window of size 16. We train the amodal expander on the TAO-Amodal training set, along with PasteNOcclude (PnO) and augmentation used in GTR (Zhou et al., 2022b). All the modules except the amodal expander are frozen during training. More ablation studies, hyperparameter details for training and PnO can be found in the appendix.

Table 4: **Multi-frame-aware amodal baselines on TAO-Amodal validation set.** We explore extensions to include multi-frame signals for fine-tuned expander. Following (Khurana et al., 2021), we use a Kalman filter to predict the positions of occluded objects, augmented by a monocular depth estimator to filter out spurious predictions. This leads to an increase in $AP^{[0,0.1]}$. Further, we integrate multi-frame cross-attended Re-ID features, feeding them into the expander with concatenation. This boosts tracking and out-of-frame metrics.

| Method | Detection Metrics | | | | | Tracking Metrics | |
|---|---|---|---|---|---|---|---|
| | $AP^{[0,0.1]}$ | $AP^{[0.1,0.8]}$ | $AP^{[0.8,1]}$ | $AP^{OoF}$ | AP | AP | $AP^{[0,0.8]}$ |
| Baseline (GTR (Zhou et al., 2022b)) | 0.8 | 13.2 | 37.5 | 14.2 | 28.2 | 16.0 | 8.9 |
| Amodal Expander | 0.8 | **16.4** (+3.2) | **37.7** (+0.2) | 17.6 | **29.9** (+1.7) | 16.4 | 10.1 |
|   + Kalman filter | 1.8 | 15.8 | 36.3 | 16.4 | 29.0 | 16.0 | 10.1 |
|   + Depth (Khurana et al., 2021) | **2.0** (+1.2) | 16.1 | 36.8 | 16.8 | 29.4 | 15.9 | 10.0 |
| Amodal Expander + Temporal Re-ID | 0.7 | 16.2 | 37.7 | **17.8** (+3.6) | 29.8 | **17.1** (+1.1) | **11.0** (+2.1) |

**Explore fine-tuning strategies for amodal perception.** We explored several fine-tuning strategies including amodal expander on TAO-Amodal validation set as shown in Tab. 3. Amodal expander trained with PnO for 45k iterations achieves 3.29% and 3.47% performance win under partially occluded ($AP^{[0.1,0.8]}$) and out-of-frame ($AP^{OoF}$) scenario. Fine-tuning entire model or solely the regression head and proposal network results in performance degradation. We posit that, with only 500 amodal training sequences, the models struggle to completely *discard* modal knowledge. Fine-tuning box regression head is suboptimal when compared to amodal expander. Amodal expander further provides flexibility to adjust the architecture and select different input information, which are both important as shown in the ablation provided in the appendix.

**Integrating temporal signals into amodal baselines.** In Tab. 4, we present two strategies for using multi-frame information within the amodal expander: 1) using a Kalman filter to forecast occluded object locations, with a monocular depth estimator to filter erroneous predictions, following (Khurana et al., 2021), and 2) incorporating temporal Re-ID features. Note that (1) can associate single-frame detections, while also predicting *new* boxes when an object is completely occluded. This significantly improves $AP^{[0,0.1]}$. For (2), we take multi-frame Re-ID features and feed them into the amodal expander with channel concatenation. This helps improve out-of-frame and tracking metrics. We discuss other avenues to integrate temporal signals into amodal trackers in Sec. D.

## 6 DISCUSSION AND LIMITATIONS

In this work, we focus on amodal perception of real-world objects. We draw inspiration from cognitive functions of amodal completion and object permanence that humans develop at an early age. Despite this, advancements in perception stacks (like object detection and tracking) do not focus on amodal understanding. To remedy this, we make three central contributions. First, we contribute a benchmark that annotates 833 categories of objects amodally in unconstrained indoor and outdoor settings, under partial and complete occlusion. Second, we contribute a benchmarking protocol in the form of metrics that evaluate detection and tracking specifically for the cases of partial or complete occlusions. Our key finding is that existing algorithms struggle under extreme occlusions, even when given access to neighboring frames that may contain less occlusion. Finally, we investigate data augmentation and fine-tuning strategies that modestly improve existing detectors and trackers. We hope our benchmark will spur further work in this important but underexplored area.

**Limitations** TAO-Amodal inevitably inherits the limitations of the TAO benchmark on which it is based. This includes TAO's low-frequency 1FPS annotations and the federated annotation protocol. Additionally, since there is an uncertainty in localizing occluded objects, our proposed AP metric with a lower threshold is not the ideal solution as it still expects methods to output a semi-precise bounding box. We discuss these limitations in more detail in the appendix.

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

# Appendix

# Contents

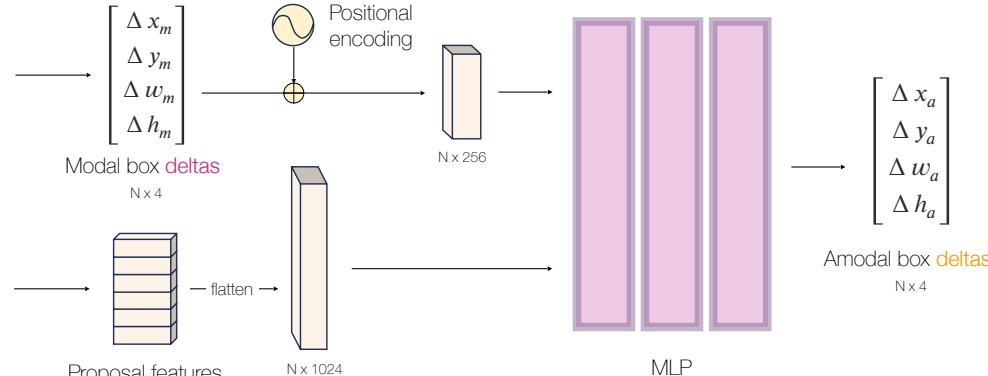

Figure 4: **Amodal Expander Architecture.** Given $N$ flattened proposal features and modal box (delta) predictions represented with 256-dim positional encodings (Vaswani et al., 2017), we predict amodal box (deltas) with a two-layer MLP (unless otherwise specified). Further architecture details are in Appendix A.

In this appendix, we extend our discussion of the proposed dataset and method within the context of tracking any object with amodal perception. We also provide a comprehensive video demonstration of our dataset and qualitative results at `webpage/index.html`.

## A IMPLEMENTATION DETAILS

### A.1 TRAINING AMODAL EXPANDER

We illustrate the amodal expander in Fig. 4. We trained amodal expander on TAO-Amodal training set for 20k iterations for all experiments unless specified. We used a 2-layer MLP as the architecture. The hidden dimension of MLP is 256. We apply ReLU (Agarap, 2018) and dropout (Srivastava et al., 2014) with a probability of 0.2 to each layer except the last one. We implemented the expander in conjunction with GTR (Zhou et al., 2022b). Architecture details of GTR align with the selection in the prior work (Zhou et al., 2022b). We used 0.01 as the base learning rate and applied `WarmupCosineLR` (He et al., 2019) as the scheduler. The optimizer is AdamW (Loshchilov & Hutter, 2017). The batch size for training is 4. We adopted the training methodology outlined in (Zhou et al., 2022b), treating each image as an independent sequence. We applied data augmentation (Zhou et al., 2020), including random cropping and resizing, to each image to produce synthetic videos with a length of 8 frames. Beyond this, we further applied PasteNOcclude, introduced in Sec. 4.3 in the main paper, on top of the synthetic videos to automatically generate more occlusion scenarios. We provide the hyperparameter details of PasteNOcclude in the next section. We utilize 4 NVIDIA GeForce RTX 3090 GPUs to train the amodal expander, a process that takes approximately 10 hours for 45k iterations. We run inference on the validation set using a single NVIDIA GeForce RTX 3090 and the process takes about 3 hours.

### A.2 PASTENOCCLUDE (PNO)

We illustrated visual examples of PnO in Fig. 6. We mask the background area with the segmentation mask and collect the cropped object from LVIS (Gupta et al., 2019) and COCO (Lin et al., 2014) to serve as occluders. We filter out segments where the mask area is less than 70% of the bounding box area to ensure that the occluder is not occluded. In the training process, we view each image as a sequence and create an 8-frame sequence employing the data augmentation strategy in GTR (Zhou et al., 2022b) based on each image. Subsequently, we randomly select 1 to 7 segments from the collection and place them at random locations. Further, we randomly adjust the height and width of the inserted segments within the range of $[12, 192]$. We randomly determine the object's location and size only in the first and last frames to ensure smooth transitions between consecutive frames. The size and location in intermediate frames are obtained through interpolation.

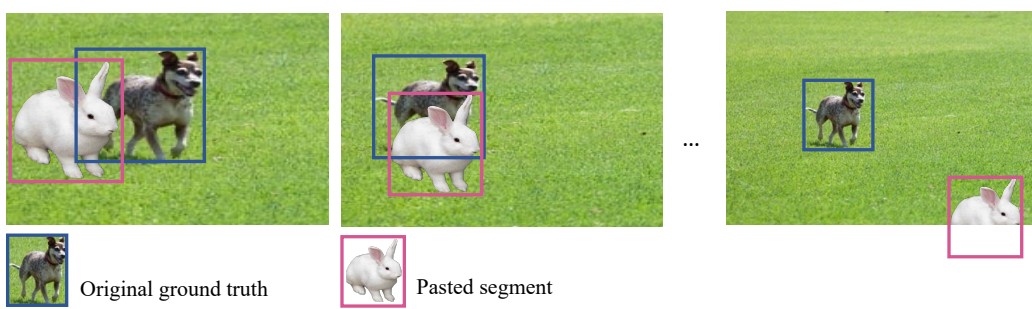

Figure 5: **Synthetic occlusions with PasteNOcclude (PnO).** PnO allows us to manually simulate occlusion scenarios and out-of-frame scenarios. We randomly choose 1 to 7 segments from a collection sourced from LVIS (Gupta et al., 2019) and COCO (Lin et al., 2014) for pasting. For each inserted segment, we randomly determine the object's size and position in the first and last frames. The size and location of the segment in intermediate frames are then generated through linear interpolation.

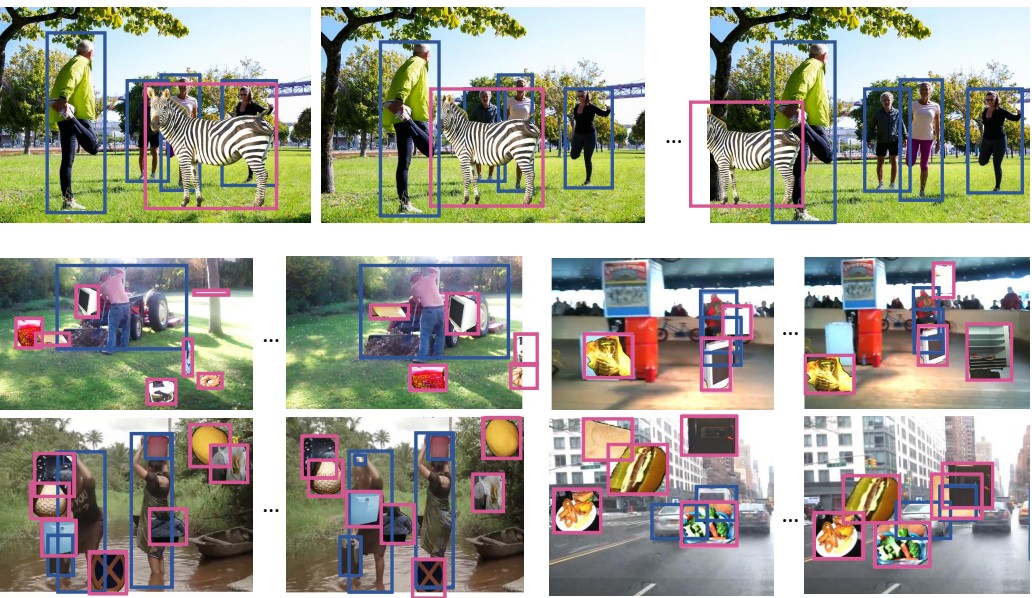

Figure 6: **More examples of PasteNOcclude (PnO).**

## B  MORE EMPIRICAL ANALYSIS

We used the evaluation metrics defined in Sec. 5.2 in the main draft. We summarize all the definitions in Tab. 5. We presented additional experiments involving state-of-the-art trackers in Appendix B.1.3 and the amodal expander in Appendix B.2.

### B.1  BENCHMARKING OFF-THE-SHELF-TRACKERS

#### B.1.1  EVALUATION ON TAO-AMODAL VALIDATION SET

We report detection and tracking average precision (AP) numbers of SOTA off-the-shelf trackers on TAO-Amodal validation set running at 1fps with an IoU threshold 0.5 in Tab. 6. We also observed similar performance trends when running at 5fps with higher IoU thresholds, shown in Tabs. 7

Table 5: **Evaluation metrics with IoU threshold 0.5.** We define variations of AP (Lin et al., 2014) and Track-AP (Dave et al., 2020a) based on levels of occlusion.

| Metric | Definition | Type |
|---|---|---|
| AP | Average Precision (AP) averaged across all categories at an IoU threshold 0.5. | |
| $AP^{[0, 0.1]}$ | AP for heavily occluded objects, with visibility smaller than 0.1. | |
| $AP^{[0.1, 0.8]}$ | AP for partially occluded objects, with visibility in [0.1, 0.8]. | Detection Metrics |
| $AP^{[0.8, 1.0]}$ | AP for non-occluded objects, with visibility larger than 0.8. | |
| $AP^{OoF}$ | AP for partially out-of-frame (OoF) objects. | |
| Modal AP | AP on modal annotations. | |
| Track-AP (Dave et al., 2020a) | Average Precision of a track averaged across all categories at an 3D IoU threshold 0.5. | |
| $Track\text{-}AP^{[0, 0.8]}$ | Track-AP for any track that is occluded, with visibility at or below 0.8, for more than 5 frames (seconds). | Tracking Metrics |
| Modal Track-AP | Track-AP on modal annotations | |

Table 6: **Off-the-shelf trackers on TAO-Amodal validation set.** Off-the-shelf trackers were either trained on TAO (Dave et al., 2020a) or on synthetic videos (Zhou et al., 2022b) generated using LVIS images (Gupta et al., 2019), with categories aligned with our dataset. While certain trackers can detect non-occluded objects well (over 35% AP), objects that are highly occluded, partially occluded, and out-of-frame remain challenging, highlighting the difference between modal and amodal tracking. We run all existing trackers at 1 fps and average AP across all categories with an IoU threshold of 0.5.

| Method | Detection Metrics | | | | | | Tracking Metrics | | |
|---|---|---|---|---|---|---|---|---|---|
| | $AP^{[0,0.1]}$ | $AP^{[0.1,0.8]}$ | $AP^{[0.8,1]}$ | $AP^{OoF}$ | Modal AP | AP | AP | $AP^{[0,0.8]}$ | Modal AP |
| QDTrack (Fischer et al., 2023) | 0.39 | 7.79 | 21.70 | 7.88 | 20.07 | 15.47 | 7.84 | 4.03 | 11.36 |
| TET (Li et al., 2022a) | 0.70 | 8.89 | 29.96 | 8.66 | 29.42 | 22.04 | 4.72 | 3.32 | 7.7 |
| AOA (Du et al., 2021) | 0.56 | 6.32 | 24.14 | 6.53 | 23.27 | 17.76 | 13.63 | 6.63 | 21.18 |
| Detic + SORT (Zhou et al., 2022a; Bewley et al., 2016) | 0.38 | 6.68 | 21.31 | 8.09 | 18.84 | 15.32 | 6.18 | 3.81 | 8.16 |
| ViTDet-B + SORT (Li et al., 2022b; Bewley et al., 2016) | 0.77 | 11.40 | 34.03 | 12.98 | 32.67 | 25.15 | 6.95 | 4.10 | 11.57 |
| ViTDet-L + SORT (Li et al., 2022b; Bewley et al., 2016) | **1.18** | 13.75 | 37.41 | 14.70 | 36.65 | 28.05 | 8.19 | 5.14 | 13.73 |
| ViTDet-H + SORT (Li et al., 2022b; Bewley et al., 2016) | 1.03 | **14.54** | **39.71** | **16.53** | **38.05** | **29.56** | 8.94 | 5.76 | 14.55 |
| GTR (Zhou et al., 2022b) | 0.78 | 13.24 | 37.54 | 14.18 | 36.08 | 28.19 | **16.02** | **8.86** | **22.50** |

and 8. Every off-the-shelf tracker was trained on either TAO (Dave et al., 2020a) or LVIS (Gupta et al., 2019), ensuring alignment of category vocabulary with our dataset as detailed in Sec. 5.2. We reproduced AOA (Du et al., 2021) using their released implementation, with object detector trained on LVIS and tracking ReID head trained on TAO.

### B.1.2 USING OFF-THE-SHELF MODAL TRACKERS FOR AMODAL PERCEPTION.

Tab. 6 reveals notable differences in detection AP between modal (Modal AP) and amodal annotations (AP), amounting to an 8.49% difference. Additionally, the amodal tracking AP experiences a substantial decline compared to modal tracking AP. These results highlight the difference between amodal and modal perception. Additionally, we found that standard trackers performing well on TAO (Modal APs) in Tab. 6 also achieve better results on TAO-Amodal in Tab. 2. This highlights a strong correlation between open-world tracking performance and success on our dataset.

### B.1.3 HOW WELL DO STANDARD TRACKERS HANDLE OCCLUSION?

Existing off-the-shelf trackers exhibit reasonable performance in detecting non-occluded objects, with ViTDet achieving 39.71% $AP^{[0.8,1]}$ as revealed in Tab. 6. However, all trackers face challenges in handling heavily occluded, partially occluded ($AP^{[0.1,0.8]}$) and out-of-frame (OoF) scenarios. We noticed that ViTDet operating at 5 fps benefits from the property of SORT to estimate the location in the current frame using past information in Tab. 8. Nevertheless, this improvement comes at the cost of processing ViT-Det on 5x more frames than models running at 1 fps. In contrast, amodal completion could be a promising way for efficiently handling occlusion.

**Evaluation with higher IoU thresholds.** In Tab. 7, we evaluate the trackers with average precision (AP) averaged over 10 IoU thresholds from 0.5 to 0.95 at a step 0.05. The performance trend basi-

Table 7: **Off-the-shelf trackers on TAO-Amodal validation with higher IoU thresholds.** The definitions of our evaluation metrics can be found in Tab. 5. The AP numbers are averaged over 10 IoU values from 0.5 to 0.95 with a 0.05 step, denoted as $AP_{0.5:0.95}$. We observed a similar performance trend as results evaluated with an IoU threshold 0.5. We run all trackers at 1 fps.

| Method | Detection $AP_{0.5:0.95}$ | | | | | | Tracking $AP_{0.5:0.95}$ | |
| --- | --- | --- | --- | --- | --- | --- | --- | --- |
| | $AP^{[0,0.1]}$ | $AP^{[0.1,0.8]}$ | $AP^{[0.8,1]}$ | $AP^{OoF}$ | Modal AP | AP | AP | $AP^{[0,0.8]}$ |
| QDTrack (Fischer et al., 2023) | 0.12 | 2.29 | 13.03 | 2.90 | 12.64 | 8.53 | 3.36 | 1.52 |
| TET (Li et al., 2022a) | 0.21 | 2.71 | 17.27 | 3.14 | 17.58 | 11.80 | 1.99 | 1.14 |
| AOA (Du et al., 2021) | 0.26 | 1.87 | 15.98 | 2.84 | 16.36 | 10.52 | 6.59 | 2.07 |
| ViTDet-B + SORT (Li et al., 2022b; Bewley et al., 2016) | 0.33 | 3.41 | 19.67 | 5.02 | 19.83 | 13.39 | 3.03 | 1.40 |
| ViTDet-L + SORT (Li et al., 2022b; Bewley et al., 2016) | **0.43** | 4.14 | 22.08 | 5.81 | 22.65 | 15.35 | 4.16 | 1.84 |
| ViTDet-H + SORT (Li et al., 2022b; Bewley et al., 2016) | 0.36 | 4.38 | 23.62 | **6.67** | 23.89 | 16.21 | 4.24 | 1.94 |
| GTR (Zhou et al., 2022b) | 0.24 | **4.60** | **26.01** | 6.62 | **26.83** | **18.07** | **7.52** | **3.05** |

Table 8: **Off-the-shelf trackers on TAO-Amodal validation set running at 5 fps.** ViTDet (Li et al., 2022b) achieves a performance gain by running at a higher fps as SORT (Bewley et al., 2016) leverages its capability to estimate the new location based on the location in previous frames. AP numbers are averaged across all categories at an IoU threshold 0.5.

| Method | Detection AP | | | | | | Tracking AP | | |
| --- | --- | --- | --- | --- | --- | --- | --- | --- | --- |
| | $AP^{[0,0.1]}$ | $AP^{[0.1,0.8]}$ | $AP^{[0.8,1]}$ | $AP^{OoF}$ | Modal AP | AP | AP | $AP^{[0,0.8]}$ | Modal AP |
| QDTrack (Fischer et al., 2023) | 0.42 | 7.59 | 21.53 | 7.78 | 19.98 | 15.42 | 6.63 | 2.72 | 10.34 |
| TET (Li et al., 2022a) | 0.24 | 5.39 | 14.56 | 4.73 | 29.42 | 10.51 | 3.52 | 2.21 | 5.56 |
| AOA (Du et al., 2021) | 0.56 | 6.29 | 24.35 | 6.77 | 23.51 | 17.85 | 12.82 | 5.53 | 20.67 |
| ViTDet-B + SORT (Li et al., 2022b; Bewley et al., 2016) | 1.00 | 13.38 | 37.98 | 14.78 | 37.08 | 28.32 | 10.09 | 4.40 | 16.93 |
| ViTDet-L + SORT (Li et al., 2022b; Bewley et al., 2016) | **1.32** | 16.38 | 43.30 | 17.16 | 42.31 | 32.08 | 11.75 | 5.53 | 19.22 |
| ViTDet-H + SORT (Li et al., 2022b; Bewley et al., 2016) | 1.06 | **17.24** | **45.18** | **18.58** | **44.02** | **33.53** | 13.16 | 5.87 | **21.39** |
| GTR (Zhou et al., 2022b) | 0.57 | 12.45 | 35.89 | 13.63 | 34.92 | 27.28 | **13.70** | **7.02** | 20.09 |

cally aligns with what we observed in Tab. 6 in the main paper. GTR (Zhou et al., 2022b) obtained strong performance in both detection and tracking. When evaluated with higher IoU thresholds, ViTDet (Li et al., 2022b) and SORT (Bewley et al., 2016) demonstrate inferior detection performance compared to GTR, indicating a contrasting outcome compared to the results obtained at a 0.5 threshold. This shows the limitations of SORT (Bewley et al., 2016) in accurately estimating bounding boxes.

**Running trackers at higher fps.** We reported the performance of state-of-the-art trackers running at 5 fps in Tab. 8. We noticed that ViTDet (Li et al., 2022b) along with SORT (Bewley et al., 2016) achieved the best performance among all the trackers. This aligns with our intuition as SORT estimates the location in the current frame based on prior-frame locations. This property benefits from running at higher fps, but it requires processing ViTDet on 5×more frames than models operating at 1 fps, heavily increasing computational demands.

## B.2 AMODAL EXPANDER EXPERIMENTS

### B.2.1 SCALING UP TRAINING DATA

In Tab. 9, we scale up the training data to 4x by including test videos as train set and evaluate the amodal expander on the validation set. We note that simply increasing the size of the training data does not significantly improve the metrics of amodal expander compared to results shown in Tab. 3.

Increasing training data improves the performance of full model fine-tuning and could be a promising direction for future work. However, even 4x more training data is insufficient to fine-tune the entire model. A more effective strategy is to freeze the modal trackers and apply an additional "correction" module (*e.g.*, amodal expander). These analyses validate our design to propose TAO-Amodal as an evaluation benchmark.

### B.2.2 DETECTING PEOPLE WITH AMODAL EXPANDER

In Tab. 10, we study how well the expander baseline detects and tracks people, which serves as a crucial category in many autonomous driving and tracking benchmarks. Amodal expander obtains a significant improvement compared to the modal baseline, particularly on $AP^{[0.1, 0.8]}$ and $AP^{OOF}$.

Table 9: **Scaling up training data for amodal expander.** All fine-tuning is done on a set of 1,928 videos, vs. 500 in the main paper.

| Method | Detection Metrics | | | | | Tracking Metrics | |
|---|---|---|---|---|---|---|---|
| | $AP^{[0,0.1]}$ | $AP^{[0.1,0.8]}$ | $AP^{[0.8,1]}$ | $AP^{OoF}$ | AP | AP | $AP^{[0,0.8]}$ |
| Baseline (GTR [59]) | 0.8 | 13.2 | 37.5 | 14.2 | 28.2 | 16.0 | 8.9 |
| Fine-tune entire model | **1.1** | 12.7 | 29.1 | 12.4 | 22.5 | 9.7 | 6.2 |
| Fine-tune regression head | 0.9 | 14.4 | **38.0** | 15.4 | 29.1 | **16.9** | 9.5 |
| Amodal Expander | 0.8 | **16.9** (+3.7) | 37.7 | **17.9** (+3.7) | **30.0** (+1.8) | 16.5 | 10.7 |
| Amodal Expander + PnO | 0.7 | 16.5 | 37.8 | **17.9** (+3.7) | **30.0** (+1.8) | 16.5 | **10.8** (+1.9) |

Table 10: **Evaluating the 'people' category.** We follow the conventions of Tab. 3 but evaluate performance only on the people category. Fine-tuned expander shows improvements over modal baseline, which can be observed in Fig. 7. We posit that this dramatic performance increase comes from the fact that people is the most common category. PasteNOcclude (PnO) leads to a slight drop for this category, which suggests that adding synthetic (occluded) examples is more helpful for less common categories.

| Method | Detection Metrics | | | | | Tracking Metrics | |
|---|---|---|---|---|---|---|---|
| | $AP^{[0,0.1]}$ | $AP^{[0.1,0.8]}$ | $AP^{[0.8,1]}$ | $AP^{OoF}$ | Overall | Overall | $AP^{[0,0.8]}$ |
| GTR (Zhou et al., 2022b) | 0.29 | 37.15 | 71.49 | 42.07 | 53.81 | 17.47 | 14.39 |
| FT regression head | 0.41 | 49.32 | 78.93 | 53.26 | 61.36 | 20.44 | 18.74 |
| Amodal Expander | 2.26 | 71.64 | 84.07 | 73.74 | 74.22 | **26.77** (+9.30) | 28.94 |
| Amodal Expander† | **2.46** (+2.17) | **71.86** (+34.71) | **84.21** (+12.72) | **73.96** (+31.89) | **74.34** (+20.53) | 26.72 | **28.95** (+14.56) |
| Amodal Expander + PnO | 1.94 | 69.87 | 83.86 | 72.58 | 73.20 | 26.68 | 28.76 |
| Amodal Expander + PnO† | 1.99 | 70.23 | 84.00 | 72.85 | 73.38 | 26.61 | 28.64 |

Tracking on highly or partially occluded people (Track-$AP^{[0.0,0.8]}$) also increases by 14.6%. This shows that one can obtain an effective amodal people tracker that could also track objects of diverse category vocabulary with our dataset using a simple fine-tuning scheme.

### B.2.3  IMPORTANCE OF PROPOSAL MATCHING STRATEGIES

To apply regression loss, training a box prediction head requires matching each region proposal to a ground truth box. A naive strategy is to directly match the region proposals to the amodal ground truth box. However, direct matching with amodal boxes leads to suboptimal results as shown in Tab. 11. As standard trackers generate modal region proposals, the model faced challenges in aligning proposals with the accurate ground truth due to a low Intersection over Union (IoU) between modal proposals and amodal ground truth. Matching proposals with modal boxes and applying regression loss using amodal ground truth yield better results.

### B.2.4  INVESTIGATING KEY INFORMATION FOR AMODAL BOX INFERENCE

Tab. 12 reports different input choices to the amodal expander. Modal box (deltas) $\Delta b$, output by the regression head as shown in Fig. 3 in the main paper, are used to yield final modal box predictions when applied to region proposals and thus contain information about the exact location of modal box predictions. Proposal features includes visual appearance information of the detected region proposals. Absence of visual cues significantly diminishes the performance of both detection and tracking under occlusion. Interestingly, the amodal expander, incorporating both modal delta and proposal features, yielded the most favorable outcomes. This indicates that estimating modal box locations also contributes to effective amodal reasoning.

### B.2.5  NUMBER OF MLP LAYERS

We tested with the depth of amodal expander architecture in Tab. 13. We observe a reverse-U pattern concerning the number of MLP layers, with two-layer MLPs demonstrating superior performance compared to other models. A one-layer MLP proves suboptimal in both detection and tracking. Notably, using a 1-layer MLP results in slightly inferior outcomes compared to fine-tuning the regression head, as indicated in Tab. 3 in the main paper. We argue that the regression head may derive benefits from pre-training on modal benchmarks.

Table 11: **Ablation: Region proposal matching strategy.** Given that modal trackers generate modal proposals, an improved strategy involves matching region proposals with modal ground truth (GT) while applying regression loss to amodal predictions against the amodal GT. Both expander models are trained with Paste-and-Occlude (PnO) on TAO-Amodal training set for 20k iterations.

| Matching | Detection AP | | | Tracking AP | |
| --- | --- | --- | --- | --- | --- |
| | $\text{AP}^{[0.1,0.8]}$ | $\text{AP}^{\text{OoF}}$ | AP | AP | $\text{AP}^{[0.1,0.8]}$ |
| Modal GT | 13.96 | 14.92 | 28.64 | **16.45** | 8.96 |
| Amodal GT | **16.41** | **17.64** | **29.87** | 16.35 | **10.13** |

Table 12: **Input to Amodal Expander.** Modal box (deltas) $\Delta b$, output by the regression head as shown in Fig. 3, contains information about the exact location of modal box predictions. Object features $f$ are embedded with visual appearance information of the modal proposals. We found that both information are important in amodally inferring the object's shape. All models were trained on TAO-Amodal training set with PasteNOcclude (PnO) for 20k iterations.

| Method | Detection AP | | | Tracking AP | |
| --- | --- | --- | --- | --- | --- |
| | $\text{AP}^{[0.1,0.8]}$ | $\text{AP}^{\text{OoF}}$ | AP | AP | $\text{AP}^{[0,0.8]}$ |
| $\Delta b$ | 13.86 | 14.79 | 28.62 | **16.47** | 8.94 |
| $f$ | 16.12 | 17.08 | 29.58 | 16.12 | 10.08 |
| $f$ and $\Delta b$ | **16.41** | **17.64** | **29.87** | 16.35 | **10.13** |

Table 13: **Number of MLP layers in Amodal Expander.** Empirically, a lightweight 2-layer MLP amodal expander is sufficient to generate reasonable amodal predictions. All models were trained on TAO-Amodal training set for 20k iterations.

| # layers | Detection AP | | | Tracking AP | |
| --- | --- | --- | --- | --- | --- |
| | $\text{AP}^{[0.1,0.8]}$ | $\text{AP}^{\text{OoF}}$ | AP | AP | $\text{AP}^{[0,0.8]}$ |
| 1-layer | 13.78 | 15.19 | 28.21 | 14.29 | 8.12 |
| 2-layer | **16.41** | **17.64** | **29.87** | **16.35** | **10.13** |
| 4-layer | 15.55 | 17.02 | 29.41 | 16.35 | 9.99 |
| 6-layer | 14.55 | 15.64 | 28.79 | 16.05 | 9.09 |

### B.3 QUALITATIVE RESULTS

We illustrate the qualitative results of amodal expander on TAO-Amodal validation set in Figs. 7 and 8. Amodal expander infers objects that are occluded under various scenarios and completes occluded objects of diverse categories.

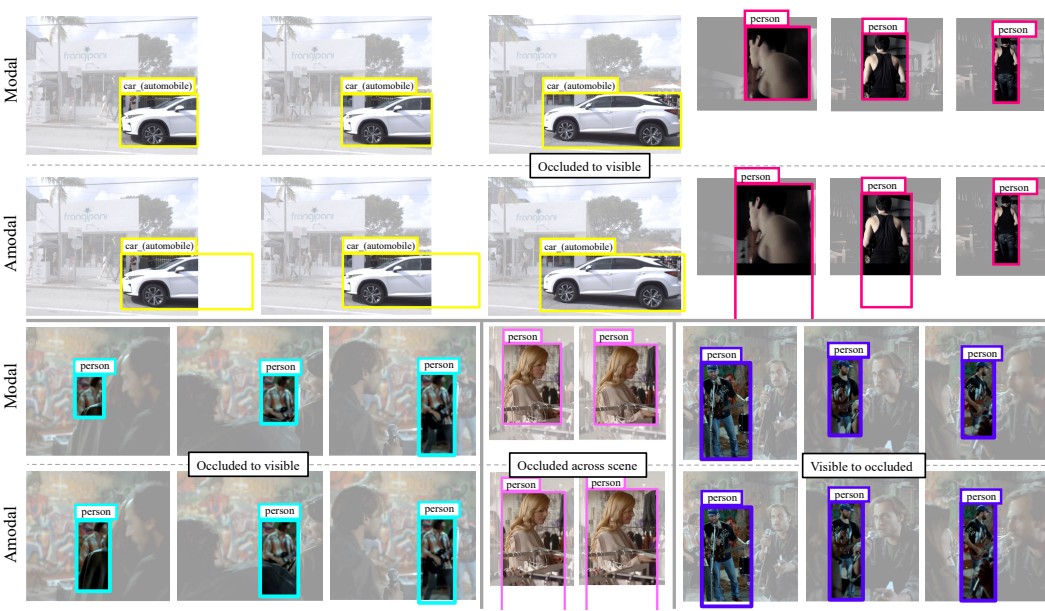

Figure 7: **Qualitative results of Amodal Expander on TAO-Amodal val.** Trackers fine-tuned with expander produce both modal and amodal predictions. The expander amodally complete objects that are occluded by objects in the scene (bottom-left) or objects that lie partially out of frame. We further verify that fine-tuned expander can amodally complete objects that were occluded in the past as well as objects that become occluded later.

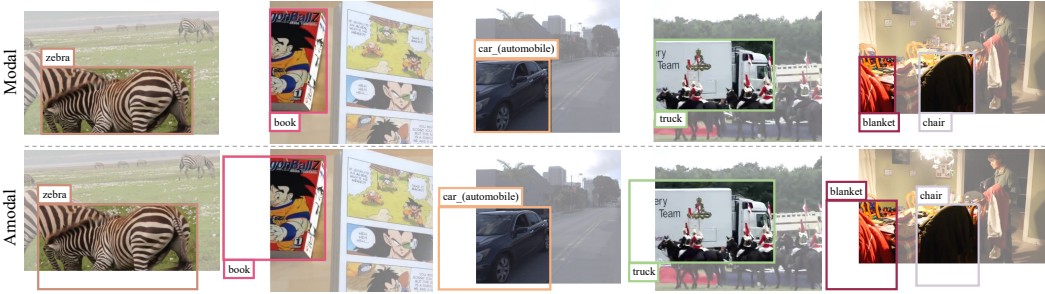

Figure 8: **Qualitative results of Amodal Expander across diverse categories on TAO-Amodal val.** Though we achieve the most impressive results for people, our Amodal Expander is effective across a diverse set of categories.

Table 14: **Annotation guidelines.** TAO-Amodal is annotated with the guidelines below, which taxonomizes occlusions across severity (partial versus complete) and type (in/out-of-frame). As mentioned in Sec. 3 in the main paper, we scope out the case where an object may be present behind the camera. For out-of-frame occlusions, we limit the *annotation workspace* to be twice the image size.

| Occlusion type | Extent | Cases | Instructions |
|---|---|---|---|
| In-frame | Partial | Partially occluded before being fully visible | Annotate with best estimate using category label |
| | | Partially occluded after being fully visible | Annotate with best estimate |
| | Complete | Invisible before being (partially) visible | Only annotate if the object has been visible before |
| | | Invisible after being (partially) visible | If confident, annotate with best estimate |
| | | | If not, only annotate till the last visible frame |
| | | Invisible for a while | If confident, annotate with best estimate |
| | | | If not, still annotate but add an uncertainty flag |
| Out-of-frame | Partial | Object goes beyond image border | Only annotate inside the annotation workspace |
| | | Object goes beyond the padded image | Clip at the border of the padded image |
| | Complete | - | - |
| Behind-the-frame | Partial | Object is in front of and behind the camera | Only label the part of object in front of camera |
| | Complete | - | - |

# C  TAO-AMODAL ANNOTATIONS

## C.1  ANNOTATION GUIDELINES

We ensure high-quality annotations by requiring annotators to follow the guidelines detailed in Tab. 14. Our coverage spans various occlusion scenarios, encompassing in-frame, out-of-frame, or behind-the-scene situations, where an object may be partially obscured behind the camera.

## C.2  OTHER ANNOTATION STATISTICS

We present the class and object occlusion distributions in Figs. 9 and 10.

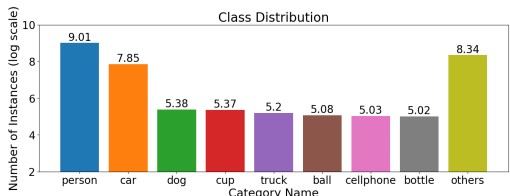

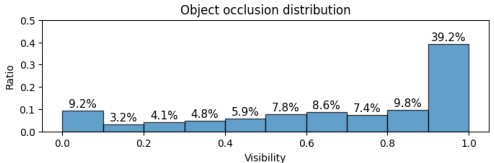

Figure 9: **Class distribution.** We present counts of instances from top 8 most frequent categories and other categories, using a logarithmic scale.

Figure 10: **Object occlusion distribution.** We plot the distribution at a 10% visibility span.

## C.3  UNCERTAIN OBJECTS

We provide examples of objects marked with an `is_uncertain` flag in Fig. 11.

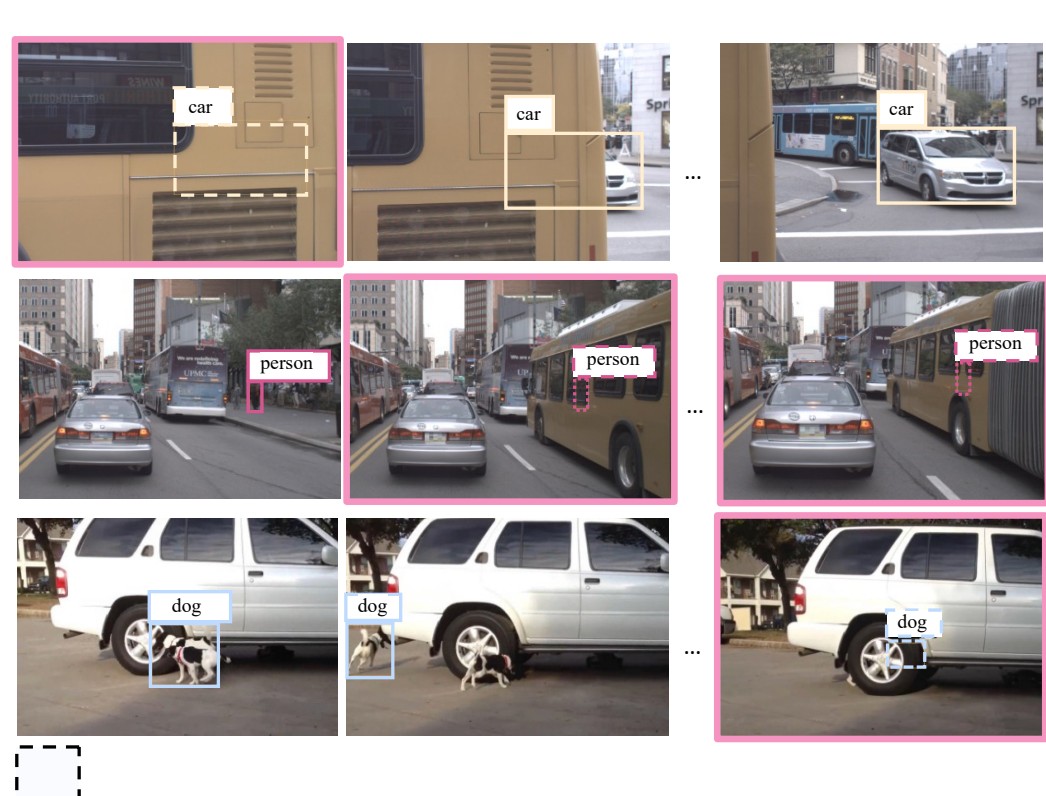

uncertain annotations

Figure 11: **Examples with *is_uncertain* flag.** Bounding boxes with dashed lines represent objects flagged with *is_uncertain*. For frames containing uncertainly annotated objects, we use pink borders. As detailed in Sec. 3, our approach ensures that over 99% of annotations are of high quality by instructing annotators to reference preceding and subsequent frames when dealing with heavily occluded objects. In cases where the annotator remains uncertain, such as with fully occluded objects or when objects do not reappear post-occlusion, they are directed to mark these objects accordingly.

# D    LIMITATIONS

Although TAO-Amodal provides a comprehensive evaluation, we recognize that the current metrics may have limitations in evaluating amodal tracking. The shape and position of heavily occluded and invisible objects are inherently uncertain, leading to multiple plausible predictions that human reviewers might deem acceptable. We could use Top-k metrics (Khurana et al., 2021) as an alternative. Yet, top-k metrics would require trackers to output a "distribution" of locations and thus make existing literature of modal trackers unsuitable for amodal tracking. Therefore, we adopt an AP threshold of 0.5 to appropriately penalize predictions of highly occluded objects. Another way we try to curb the inherent uncertainty for amodal evaluation is to ensure precise amodal annotations by referring neighboring frames and through multiple rounds of quality check, which we discussed in Sec. 3 of the main paper.

TAO-Amodal also inevitably inherits the limitations of the TAO benchmark on which it is based. This includes TAO's low-frequency 1FPS annotations and the federated annotation protocol. We trade off low-frequency annotation with high-quality annotations given a limited budget of human labor. We follow federated annotation protocol because it is widely used in image recognition (Gupta et al., 2019) and multi-object tracking (Dave et al., 2020a) *benchmarks* with extensive vocabularies, where exhaustive labeling is too expensive. Lastly, we observe that the detection performance improvements of the fine-tuned amodal expander are modest compared to ViTDet (Li et al., 2022b). Our amodal expander is built upon GTR (Zhou et al., 2022b) following the concept of lightweight instruction tuning. Drawing parallels to the success in NLP, we hope our empirical findings will inspire further research into more effective fine-tuning strategies for enhancing existing modal trackers to operate in the amodal domain.

Lastly, despite our exploration of temporal aware baselines in Tab. 4, we acknowledge the limitations of frame-independent detectors, a challenge shared by many current modal trackers (Zhou et al., 2022b; 2020). Typically, these trackers are trained on image datasets with more comprehensive annotations, and an additional temporal tracking module is trained on tracking datasets while the detector is frozen. This approach is to prevent performance degradation in object detection after fine-tuning on tracking data, which we observed when fine-tuning the region proposal network in Tab. 3. Therefore, building a temporal-aware detector remains challenging but presents a promising direction for advancing both amodal and modal tracking.

