# OpenReview forum: "TAO-Amodal: A Benchmark for Tracking Any Object Amodally"
_ICLR.cc/2025/Conference — ICLR 2025 Conference Withdrawn Submission_

### Official Review · Reviewer_NS5G · 2024-10-30

**Soundness:** 3
**Presentation:** 2
**Contribution:** 3
**Rating:** 8
**Confidence:** 4

**Summary:**

This paper introduces TAO-Amodal, a large-scale benchmark dataset for amodal object tracking, built upon the existing TAO dataset. The work addresses an important gap in computer vision by providing annotations for tracking objects under heavy occlusions and out-of-frame scenarios across 833 diverse categories in 2,907 videos. The authors evaluate current trackers and propose simple but effective finetuning schemes to improve amodal tracking performance.

**Strengths:**

1. The paper introduces the first large-scale amodal tracking benchmark that encompasses a diverse range of object categories, which is a significant advancement in the field. This benchmark addresses a critical gap in current datasets and tracking evaluations, as it provides a platform to assess and improve tracking algorithms under occlusion scenarios.
2. The authors have developed a comprehensive annotation protocol that includes rigorous quality control measures, ensuring the reliability of the dataset. They have thoughtfully considered both in-frame and out-of-frame occlusions, which are essential for a holistic evaluation of tracking algorithms.
3.The dataset is constructed on a large scale, with 332k boxes annotated across 2,907 videos, providing a substantial dataset for training and evaluating tracking models. It covers an impressive 833 object categories, making it one of the most diverse datasets available. The quality of the annotations is exceptionally high, with over 99% of the annotations meeting the quality standards set by the authors, which is crucial for the development and testing of robust tracking algorithms.

**Weaknesses:**

1. The training set appears to be relatively small compared to the validation and test sets. Could the authors explain the rationale behind this dataset split? It would be useful to discuss how the smaller training set might impact model performance. Additionally, considering options like data augmentation could help maximize the utility of the available training data and potentially improve model robustness.
2. A more detailed analysis of failure cases, such as full occlusion,  would be advantageous, as it would help to better understand the limitations of the models and potential areas for improvement. Including metrics or visualizations to highlight these limitations would add depth to the analysis and offer valuable insights for future research.
3.To maintain the state-of-the-art in tracking research, the authors are encouraged to incorporate the latest tracking methodologies, such as Hybrid-SORT [1] and MeMOTR [2], into the study.

[1] Yang, M., Han, G., Yan, B., Zhang, W., Qi, J., Lu, H., & Wang, D. (2023). Hybrid-SORT: Weak Cues Matter for Online Multi-Object Tracking. AAAI Conference on Artificial Intelligence.
[2] Gao, R., & Wang, L. (2023). MeMOTR: Long-Term Memory-Augmented Transformer for Multi-Object Tracking. 2023 IEEE/CVF International Conference on Computer Vision (ICCV), 9867-9876.

**Questions:**

1. Could the authors provide more detailed analysis of how performance varies across different object categories?
2. The paper would benefit from more discussion about the annotation uncertainty in heavily occluded cases and its impact on evaluation metrics.

---

> ### Author Response · Authors · 2024-11-18
> **Reply to reviewer NS5G (1/2)**
>
> Thank you for your review, we appreciate your comments that “TAO-Amodal is a significant advancement in the field”, “The comprehensive annotation protocols ensure the reliability and thoughtfully cover both in-frame and out-of-frame occlusions”, and that TAO-Amodal is “one of the most diverse datasets available”. We address your concerns below.
>
> &nbsp;
> > The training set appears to be relatively small. What is the rationale behind this dataset split?
>
> Following TAO [4], we propose TAO-Amodal primarily as an *evaluation* benchmark using a larger ‘validation’ and ‘test’ set. We construct the training set to align modal trackers
> with amodal data and propose the validation set for empirical analysis. We reserve the testing set for challenge evaluation following TAO [4].
>
> We made this design choice to ensure the evaluation quality given a limited budget of human labor. Further, our analysis on scaling training dataset size in Sec. B.2.1 suggests our training set is sufficient for aligning a standard tracker for amodal tracking.
>
> Despite the relatively small training set, a high-quality evaluation benchmark could also significantly benefit the model training. We reiterate our discussions from Sec. 3.2 below:
>
> High-quality benchmarks drive the need for innovation in curating large-scale training data. This mirrors the evolution in large language models, where the introduction of challenging benchmarks [2, 3], has led to the collection of more comprehensive training data [4]. Some benchmarks [5] do not even have a training set and rely on the use of "internet as a training set” (c.f. Sec. 5).
>
> In computer vision, we are starting to see this paradigm shift with the use of synthetic data:
> - Amodal segmentation methods [6, 7, 8] create synthetic amodal data through pasting object segments onto images
> - Prominent modal tracking methods [9, 10] generate synthetic training videos through random cropping and resizing of static image datasets [11] to compensate for the lack of large vocabulary tracking data
> - State-of-the-art methods in 3D vision (e.g., monocular depth estimation [12] and scene flow [13]) use out-of-distribution synthetic training samples, and beat prior work on real-world evaluation
>
> Additionally, we note that despite the same small training set of the original TAO dataset, the modal tracking performance has increased from 10.2 to 27.5 TrackAP50 over the years [14], which includes performance increase on object categories where little to no training data existed.
>
> Finally, we provide some insights into the potential effects of scaling amodal training data through additional analysis in *Sec. B.2.1*. We provide further discussions on our dataset splits design and its benefits in *Sec. 3.2*.
>
> &nbsp;
> > How might smaller training set impact model performance?
>
> We provide insights into the effects of scaling training data for amodal tracking through additional analysis in *Sec. B.2.1*. We note that simply increasing the size of the training data
> does not significantly improve the metrics of amodal expander compared to results shown in Tab. 3 of our paper.
>
> Increasing training data improves the performance of full model fine-tuning and could be a promising direction for future work. However, even 4x more training data is insufficient to fine-tune the entire model. This suggests that a more effective strategy is to freeze a strong modal tracker and apply an additional ”correction” module (e.g., amodal expander). These analyses further validate our design to propose TAO-Amodal as an evaluation benchmark.
>
>
> &nbsp;
> > Options like data augmentation could help maximize the utility of the available training data and potentially improve model robustness.
>
> Thanks for your suggestion. We acknowledge that data augmentation is helpful for improving model robustness and thus propose the PasteNOcclude (PnO) augmentation in this work. PnO is indeed inspired from augmentation techniques that paste objects onto images, widely used in detection and segmentation tasks. We adapted these techniques for amodal tracking to simulate occlusion scenarios, resulting in performance improvements, as shown in Table 4. This highlights that **properly adapting methods from the detection community is crucial for advancing amodal tracking**, especially given the current scarcity of training data in the amodal domain.

---

> > ### Author Response · Authors · 2024-11-18
> > **Reply to reviewer NS5G (2/2)**
> >
> > &nbsp;
> > > A more detailed analysis of failure cases, such as full occlusion, would be advantageous.
> >
> > Thanks for your comment. We did observe some interesting cases and will provide more examples in the supplement. For instance, we observed an interesting failure scenario where the amodal expander struggles to *retain* the correct bounding box size. Specifically, when a bounding box initially identifies a jersey, the expander erroneously enlarges it to include the entire human body. This issue likely arises from an annotation imbalance, where there are significantly more annotations for people than for items like shirts. Exploring strategies to address class imbalance in the amodal domain presents a promising direction for future research.
> >
> > &nbsp;
> > > The authors are encouraged to incorporate the latest tracking methodologies to maintain the state-of-the-art in tracking research.
> >
> > Thanks for your suggestion. To provide more comprehensive signals, we are happy to evaluate open-vocabulary SOTA trackers such as OVTrack [1], which is capable of tracking any object classes by using vision-language models for classifications and associations. Our team is actively working to obtain the results and will share them as soon as the evaluation is complete. We will gladly add Hybrid-SORT [2] and  MeMOTR [3] into our tables at a later stage.
> >
> >
> > [1] Zhou et al. OVTrack. CVPR 2023. \
> > [2] Yang et al. Hybrid-SORT. AAAI 2024. \
> > [3] Gao et al. MeMOTR. ICCV 2023 \
> > [4] Dave et al. TAO. ECCV 2020.

---

> > > ### Comment · Reviewer_NS5G · 2024-11-28
> > >
> > > I would like to express my gratitude to the authors for their efforts in addressing my comments and concerns. However, after considering the feedback from the other reviewers, I am unable to raise my score.

---

### Official Review · Reviewer_gZqU · 2024-10-30

**Soundness:** 2
**Presentation:** 2
**Contribution:** 2
**Rating:** 5
**Confidence:** 5

**Summary:**

The paper introduces a new benchmark dataset, TAO-Amodal, which focuses on amodal perception. This is the ability to understand the full extent of objects, even when they are partially or fully occluded. The dataset includes diverse object categories and scenarios, such as objects that are both partially or fully occluded, and those that may be out of the frame in video sequences. The authors benchmark several state-of-the-art tracking methods and introduce fine tuning strategies that improve amodal tracking performance slightly.

**Strengths:**

The paper annotated a large-scale dataset that extends the capabilities of existing modal object tracking systems to amodal scenarios.

The authors benchmark several state-of-the-art tracking methods to evaluate the dataset.

To address occlusion challenges, the paper presents a series of new evaluation metrics suitable for tracking and detecting heavily and partially occluded objects. These metrics enhance the granularity and accuracy of existing evaluation methods.

The writing of the paper is good and easy to follow.

**Weaknesses:**

1. The main contribution of the paper is building a dataset for occluded objects. However, in Table 1, NuScenes has more Occlude boxes and tracks. Besides, the annotation FPS of TAO-amodal is only 1 fps, which is much smaller than previous datasets.


2. Although this is a paper primarily contributing a dataset, it is better to propose a method related to the most critical point of this paper (occluded objects) in order to effectively convey the importance of this work. The proposed fine-tuning method is too simple. The technique innovation is limited.


3. How can the quality of the annotations in the submitted dataset be ensured? There should be some quantitative metrics to demonstrate the accuracy of the annotations.


4. The TAO-Amodal training set is relatively small, which may restrict the ability to train models from scratch and may require pretraining on modal datasets.

5. What are the primary obstacles preventing end-to-end multi-object tracking (MOT) models from achieving optimal performance on the TAO-Amodal benchmark? What factors primarily contribute to this difficulty?

6. Despite fine-tuning and data augmentation, the performance improvements (2.1% in tracking and 3.3% in detection) are modest, indicating that current systems still struggle with occlusions.

7. Another important issue is the impact of the performance on normal object detection/tracking. It would be problematic if we only obtain minor performance improvements on amodal targets, but gain worse results for normal objects.

8. The experiments conducted in the paper primarily focus on a single dataset (TAO-Amodal). Broader testing across multiple datasets could provide more compelling results.

9. In the experimental section, further Analysis of Data Annotation Uncertainty is needed. The paper does not thoroughly explore the specific impact of uncertainty on evaluation outcomes, which warrants deeper investigation.

10. The compared methods are somehow out-of-date. Thus, the experimental results are not convincing.

11. More details about the baseline are needed, such as network architecture configurations.

**Questions:**

Please see the above.

---

> ### Author Response · Authors · 2024-11-19
> **Reply to reviewer gZqU (1/5)**
>
> Thank you for your review. We address your concerns below.
>
> &nbsp;
> > `Q1:` What are the primary obstacles preventing end-to-end multi-object tracking (MOT) models from achieving optimal performance on the TAO-Amodal benchmark? What factors primarily contribute to this difficulty?
>
> Thanks for pointing out this interesting discussion. We discuss several challenges below.
>
> **Limited amodal training data.** Compared to the standard tracking, the training data available for amodal tracking remains limited, making the development of end-to-end amodal tracking models particularly challenging. Further, tracking models heavily depend on large-scale detection datasets like LVIS [1] to pre-train the detector, while a comparable detection dataset is missing in the amodal domain.
>
>
> To address the data limitations, we investigate effective fine-tuning strategies (amodal expander) and augmentation techniques (PnO) to adapt standard trackers with reduced data requirements, as the original dataset size is adequate for adapting standard trackers to amodal tracking. We believe that leveraging synthetic data for amodal training represents a promising direction for future research. Additionally, we provide analysis into the potential effects of scaling amodal training data in *Sec. B.2.1* and discuss how *our datasets might benefit model training* in Sec. 3.2.
>
>
> **The inherent limitations of model architecture for handling occlusions.**  In this work, we extend the amodal expander to incorporate temporal information (Table 4), motivated by the fact that tracking heavily occluded objects often depends on leveraging their past information.
> However, these adapted tracking models rely on *frame-independent detectors*, which limits their effectiveness in detecting heavily occluded objects. This is a challenge shared by most current trackers.
> Typically, these trackers are trained on image datasets with more comprehensive annotations, and an additional temporal tracking module is trained on tracking datasets, keeping the detector frozen. This approach avoids performance degradation in detection after fine-tuning on tracking data, which we also observed when fine-tuning the region proposal network in Tab. 3. Therefore, building a temporal-aware detector remains challenging but presents a promising direction for advancing both amodal and modal tracking.
> We provide this discussion in Sec. D of the paper.
>
> **Open-world tracking is challenging.** As TAO-Amodal is an open-world amodal tracking dataset, the model inherently encounters challenges typical of open-world scenarios, including the need to handle hundreds or even thousands of object categories with considerable variations in appearance and shape. For example, the issue of data imbalance, which is commonly observed in object detection [1] and tracking [2], poses a significant obstacle to generalization across diverse categories.
>
> Consistent with the statements above, we found that standard trackers performing well on the open-world tracking dataset [2], as shown by the Modal APs in Table 6, also achieve better results on TAO-Amodal. This highlights a strong correlation between open-world tracking performance and success on our dataset.
>
> **Challenges for adapting standard trackers into amodal trackers.** We observed interesting failure cases that may limit performance of the adapted amodal trackers on TAO-Amodal. One intriguing example involves the amodal expander failing to *retain* the correct bounding box size. Specifically, when a bounding box initially identifies a jersey, the expander erroneously enlarges it to include the entire human body. This issue likely arises from an annotation imbalance, where there are significantly more annotations for people than for items like shirts. Exploring strategies to address class imbalance in the amodal domain presents a promising direction for future research.
>
>
> Each of these challenges represents an underexplored yet promising research opportunity to push the boundaries of amodal tracking. We hope that TAO-Amodal, along with the insights provided in this work, will inspire the research community to delve deeper into this area and develop more robust approaches to amodal tracking.

---

> ### Author Response · Authors · 2024-11-19
> **Reply to reviewer gZqU (2/5)**
>
> > `Q2:` NuScenes has more occluded boxes and tracks.
>
> We appreciate the reviewer’s observation regarding the comparison between TAO-Amodal and NuScenes. Unlike NuScenes, which is optimized for autonomous driving and thus features numerous boxes for vehicles and pedestrians, TAO-Amodal is designed to evaluate open-world amodal tracking, where the emphasis is on category and scene variability rather than the density of annotations. We highlight some differences below.
>
> TAO-Amodal serves as a large-scale evaluation benchmark with a focus on **diversity**, incorporating 833 categories for a more thorough assessment. In contrast, NuScenes is narrower in scope, concentrating on pedestrians and vehicles, which are categorized into 23 classes. TAO-Amodal draws from 7 different data sources (detailed in Sec. 3) and includes a wide variety of scenes, including road scenes, indoor environments, and scenarios with complex human-object interactions. On the other hand, NuScenes is specifically geared toward autonomous driving, featuring various road scenes.
>
> &nbsp;
> > `Q3:` The annotation FPS of TAO-amodal is only 1 fps, which is much smaller than previous datasets.
>
> We provide the discussion of annotation FPS in Sec. D of the appendix. We elaborate on a few points below.
>
> Given limited annotation budget, we consciously adopted low-frequency (1 FPS), high-quality annotations to ensure the benchmark's reliability for evaluation purposes. While we can use interpolation to increase the annotation FPS between frames, we chose not to adopt this approach to maintain the integrity and accuracy of the annotations, as TAO-Amodal is positioned as a large-scale evaluation benchmark, where the focus is on quality to ensure robust evaluation.
>
>
> &nbsp;
> > `Q4:` Although this is a paper primarily contributing a dataset, it is better to propose a method related to the most critical point (occluded objects). The proposed fine-tuning method is too simple and the technique innovation is limited.
>
> We recognize that there is significant room for improvement in our methods, with the ultimate goal being the development of an end-to-end amodal tracker. However, *developing an end-to-end amodal tracker is highly challenging*, as evidenced by the performance degradation of fully fine-tuned standard trackers shown in Table 3.
>
> In return, we set the major goal in this work as developing a high-quality evaluation benchmark while additionally providing insights into effective baselines (Table 3) and extensions, such as data augmentation (PnO) and integrating amodal expander with temporal information in Table 4, to better handle occlusions in amodal tracking. We outline the motivations, design choices, and the observations behind these approaches below.
>
>
> **Amodal expander.**  We introduced the amodal expander to bridge the gap by adapting the well-established community of standard trackers for amodal tracking. We built amodal expander on top of GTR, as this method shows strong results in both detection and tracking in Tab. 2, and observed performance gains over other fine-tuning schemes in Tab. 3. Using the expander as a plug-in module upon standard trackers might thus serve as an effective baseline for future work in developing stronger amodal tracking methods.
>
>
> **Improve occlusion handlings by extending amodal expander.** To further support progress in amodal tracking, we conducted a comprehensive analysis exploring extensions of the amodal expander. For example, as shown in Table 4, we integrated temporal-aware features such as Kalman filter-based predictions, which improve detection of heavily occluded objects by utilizing prior position data. Additionally, incorporating ReID features through attention layers across multiple frames enhances tracking performance, demonstrating the potential of temporal cues.
>
>
> **PasteNOcclude.** PnO is inspired from augmentation techniques that paste objects onto images, widely used in detection and segmentation tasks. We adapted these techniques for amodal tracking to simulate occlusion scenarios, resulting in performance improvements, as shown in Table 4. This highlights that *properly adapting methods from the detection community is crucial for advancing amodal tracking*, especially given the current scarcity of training data in this domain.
>
>
> While there is room for refinement, we believe these approaches provide strong baselines and valuable insights to guide the advancement of future amodal tracking methods. Finally, we provide a comprehensive discussion on the limitations and future directions in Sec. D of the appendix.

---

> ### Author Response · Authors · 2024-11-19
> **Reply to reviewer gZqU (3/5)**
>
> > `Q5:` How can the quality of the annotations in the submitted dataset be ensured? There should be some quantitative metrics to demonstrate the accuracy.
>
> We ensure the quality of the dataset by designing detailed annotation protocols as described in Sec. 3 and Table 14.
>
> Specifically, our annotation protocol ensures the quality in several ways:
> - **Well-tailored Annotation Protocols.** We create well-defined protocols tailored to various occlusion scenarios, such as in-frame, out-of-frame occlusions. We further identify possible visibility change scenarios and establish specific guidelines. For example, when an object is invisible before reappearing, we annotate those invisible frames only if this object was previously visible (from visible to invisible). These structured annotation guidelines not only standardize the development of future amodal tracking datasets but also highlight challenging occlusion scenarios.
> - **Refer to neighboring frames and use Uncertain Flags.** We ask annotators to refer to both preceding and succeeding frames, adhering to the guidelines outlined in Table 14 for objects that are highly occluded. All occluded objects whose locations cannot be estimated from neighboring frames are marked with an `is_uncertain` flag, which account for < 1% of the data.
> - **Quality Control.** We also conduct a rigorous quality control process as detailed in Sec. 3. Specifically, each bounding box is annotated by one individual and undergoes two rounds of quality check by two separate annotators. Finally, a manual review by the authors of this work ensures the high quality of $> 99$% annotations, even for highly occluded objects. In other words, each bounding box is agreed upon by four different individuals.
>
> These efforts ensure that $> 99$% annotations align with human-level accuracy, and ensure that uncertain annotations are correctly identified using the `is_uncertain` flag. We provide some examples for objects annotated with `is_uncertain` in Fig. 11.
>
>
>
> &nbsp;
> > `Q6:` The TAO-Amodal training set is relatively small, which may restrict the ability to train models from scratch and may require pre-training on modal datasets.
>
> Following TAO [4], we propose TAO-Amodal primarily as an **evaluation** benchmark using a larger ‘validation’ and ‘test’ set. We made this design choice to ensure the evaluation quality given a limited budget of human labor. Further, our analysis on scaling training dataset size in Sec. B.2.1 suggests our training set is sufficient for aligning a standard tracker for amodal tracking.
>
> Despite the relatively small training set, a high-quality evaluation benchmark could also significantly benefit the model training. We reiterate our discussions from Sec. 3.2 below.
>
> High-quality evaluation benchmarks *drive the need for innovation in curating large-scale training data*. This mirrors the evolution in large language models, where the introduction of challenging benchmarks [2, 3], has led to the collection of more comprehensive training data [4]. Some benchmarks [5] do not even have a training set and rely on the use of "internet as a training set” (c.f. Sec. 5).
>
> In computer vision, we are starting to see this paradigm shift with the use of synthetic data:
> - Amodal segmentation methods [6, 7, 8] create synthetic amodal data through pasting object segments onto images
> - Prominent modal tracking methods [9, 10] generate synthetic training videos through random cropping and resizing of static image datasets [11] to compensate for the lack of large vocabulary tracking data
> - State-of-the-art methods in 3D vision (e.g., monocular depth estimation [12] and scene flow [13]) use out-of-distribution synthetic training samples, and beat prior work on real-world evaluation
>
> Additionally, we note that despite the same small training set of the original TAO dataset, the modal tracking performance has increased from 10.2 to 27.5 TrackAP50 over the years [14], which includes performance increase on object categories where little to no training data existed.
>
> Finally, we provide some insights into the potential effects of scaling amodal training data through additional analysis in *Sec. B.2.1*. We provide further discussions on our dataset splits design and its benefits in *Sec. 3.2*.

---

> ### Author Response · Authors · 2024-11-19
> **Reply to reviewer gZqU (4/5)**
>
> > `Q7:` Despite fine-tuning and data augmentation, the performance improvements (2.1% in tracking and 3.3% in detection) are modest, indicating that current systems still struggle with occlusions.
>
> We totally agree. Predicting occluded objects is a challenging task that people in the tracking community tried to tackle for years. This task could hardly be solved within the scope of this work. However, amodal tracking presents an interesting avenue for handling occlusions. This motivates us to initiate a solid first step by developing an amodal evaluation benchmark and effective amodal baselines to advance the community. Through this work, people will gain more insights into the following topics:
> - What are the protocols for developing amodal datasets? (Sec. 3)
> - What are some possible occlusion scenarios? (Table 14)
> - How to adapt widely available standard trackers for amodal tracking? (Sec. 4.2)
> - At which levels of occlusion does the model fail the most? (Sec. 5)
> - What is the potential direction for improving amodal baselines? (Table 4)
> - What are the limitations of existing trackers? (Sec. D)
>
> In addition to the topics we mentioned above, we also discussed the major challenges for amodal tracking in our reply to the “primary obstacles” comment. You can also find similar discussions in Sec. D of our paper.
>
>
> &nbsp;
> > `Q8:` Another important issue is the impact of the performance on normal object detection/tracking. It would be problematic if we only obtain minor performance improvements on amodal targets, but gain worse results for normal objects.
>
> Thanks for pointing this out. We can obtain this signal by comparing the results in Table 3 and Table 6, which presents results of off-the-shelf models on TAO-Amodal. The results show that the performance on visible objects ($AP^{[0.8, 1.0]}$) was maintained while results on occluded objects ($AP^{[0.1, 0.8]}$) significantly improved. We can also analyze the Modal AP and other AP metrics in Table 6 to determine the correlation between standard tracking (Modal AP) and amodal tracking (other AP metrics).
>
>
> &nbsp;
> > `Q9:` The experiments conducted in the paper primarily focus on a single dataset (TAO-Amodal). Broader testing across multiple datasets could provide more compelling results.
>
> Thank you for your suggestion. Our primary objective is to introduce TAO-Amodal as a large-scale amodal evaluation benchmark, which requires us to present comprehensive benchmarking results and analyses for this dataset within the constraints of limited pages.
>
> Nonetheless, we are pleased to include additional evaluation results on the open-world standard tracking dataset, TAO [2], as shown by the modal APs in Table 6. Our findings indicate that standard trackers excelling on TAO also perform better on TAO-Amodal, demonstrating a strong correlation between open-world tracking performance and success on our benchmark.
>
>
>
> &nbsp;
> > `Q10:` In the experimental section, further Analysis of Data Annotation Uncertainty is needed.
>
> Thanks for this suggestion. We ensure that $> 99$% annotations align with human-level accuracy and $< 1$% of the objects are annotated with an `is_uncertain` flag. We provide some examples for objects annotated with `is_uncertain` in Fig. 11. This discussion could be found in *Sec. 3*. Below, we provide further discussion on why annotations with human-level accuracy are sufficient.
>
> We posit that annotations of human-level accuracy are sufficient to drive progress in amodal tracking. For example, when a basketball is placed under a box, humans can approximate the ball's overall position but struggle to determine its exact shape and state. This underscores both the inherent difficulty of amodal tracking and the idea that expecting models to produce more precise estimations than human annotators is likely too ambitious at this point. A more practical approach would be to focus on enabling models to approximate object states in a manner comparable to human perception as an initial step in advancing amodal tracking.
>
>
>
> &nbsp;
> > `Q11:` The paper does not thoroughly explore the specific impact of uncertainty on evaluation outcomes, which warrants deeper investigation.
>
> We ensure the annotation quality to avoid uncertainty as described in the previous comment. We further discuss how we might use the evaluation metrics to curb the uncertainty of amodal tracking in Sec. 5.1 and Sec. D. We reiterate some points below:
>
> The shape and position of heavily occluded and invisible objects are inherently uncertain, leading to multiple plausible predictions that human reviewers might deem acceptable. One way to curb the effect of uncertain objects is to use Top-k [6] probabilistic metrics. However, top-k metrics require trackers to output a “distribution” of locations and thus make existing literature of modal trackers unsuitable. On the other hand, the adaptation of AP50 metrics enables us to align existing trackers for amodal tracking simply using data from TAO-Amodal.

---

> ### Author Response · Authors · 2024-11-19
> **Reply to reviewer gZqU (5/5)**
>
> > `Q12:` The methods are somehow out-of-date. Thus, the experimental results are not convincing.
>
> Thanks for the suggestion. We are happy to evaluate OVTrack [1], an open-vocabulary tracker that is capable of tracking any object classes by using vision-language models for classifications and associations. Our team is actively working to obtain the results and will share them as soon as the evaluation is complete.
>
> &nbsp;
> > `Q13:` More details about the baseline are needed, such as network architecture configurations.
>
> We are happy to add more details besides the answers below if the reviewer has additional questions.
>
> We provide training and network configurations details in Sec. A.1. Particularly, we followed the same architecture as GTR [3]. We used a 2-layer MLP as the amodal expander where the hidden dimension of MLP is 256. We apply ReLU and dropout with a probability of 0.2 to each layer except the last one.
>
> [1] Gupta et al. LVIS. CVPR 2019. \
> [2] Dave et al. TAO. ECCV 2020. \
> [3] Zhou et al. GTR. CVPR 2022. \
> [4] Zhou et al. OVTrack. CVPR 2023

---

> ### Author Response · Authors · 2024-11-25
> **Additional Evaluation Results**
>
> Dear Reviewer,
>
> We further evaluated the open-vocabulary tracker, OVTrack [1], on TAO-Amodal. We showed the results below:
>
> | Method                  | AP$^{[0, 0.1]}$ | AP$^{[0.1, 0.8]}$ | AP$^{[0.8, 1.0]}$ | AP$^{OoF}$ | AP   | Track-AP | Track-AP$^{[0.8, 1.0]}$ |
> |-------------------------|-----------|-------------|-----------|-------|------|----|-----------|
> | OVTrack [1]    |  0.88  |  12.20   |   37.08    | 12.31| 26.60   | 15.00  | 6.99 |
>
>
> &nbsp;
> &nbsp;
>
> ---
> **Analysis and insights.** OVTrack is an open-vocabulary tracker that is capable of tracking any object classes by using vision-language models for classifications and associations. Leveraging its ability to handle diverse object categories, OVTrack demonstrates strong results on visible objects. This highlights that open-vocabulary models are another interesting avenue for achieving open-world amodal tracking. OVTrack performs worse than GTR on tracking metrics for occluded objects, as shown in Table 2 of the paper, suggesting the significance of temporal cues. We hope these results could provide additional insights to inspire promising and intriguing future directions. We will update these numbers to Table 2 at the end of the discussion stage.
>
> &nbsp;
> &nbsp;
>
> [1] Zhou et al. OVTrack. CVPR 2023.

---

> ### Author Response · Authors · 2024-11-25
> **Addressed Concerns and Open to Further Discussions**
>
> Dear Reviewer,
>
> We appreciate your valuable comments. We have made a significant effort to address your concerns, including the following:
> - Outlined the primary obstacles of amodal tracking
> - Highlighted the distinct features and values of TAO-Amodal and NuScenes.
> - Provided additional insights and extensions of our methods.
> - Expanded the discussion on the annotation protocols of TAO-Amodal to clarify its benefits for future data curation.
> - Discussed how we addressed the uncertainty in amodal tracking.
> - Reiterated the rationale behind our dataset splits design.
> - Introduced additional evaluations on standard tracking, supplementary datasets, and open-vocabulary methods for comprehensive benchmarking.
>
>
> We believe that these discussions have helped us improve the clarity of our paper and provided great insights for future research avenues. We deeply appreciate your feedback and look forward to further discussions.
>
> We welcome any further feedback before the discussion deadline.
> Thank you!
>
> Best regards, Authors

---

> ### Author Response · Authors · 2024-11-26
> **Gentle Reminder by Authors**
>
> Dear Reviewer gZqU,
>
>
> We want to send you a gentle reminder that we have worked diligently to address your concerns. Please feel free to reach out if you have any further questions or require additional clarifications—we are more than happy to continue the discussion and strengthen our paper.
>
>
> Best regards,
> Authors

---

> ### Comment · Reviewer_gZqU · 2024-11-28
> **Thanks for the reply**
>
> Though some of my concerns have been clarifed, I insist that the novelty of this work cannot meet the bar of ICLR. The algorithm part is too simple and does not provide much insight into the problem itself.

---

> ### Author Response · Authors · 2024-12-02
> **Clarifications on Novelty**
>
> We appreciate the reviewer’s feedback and would like to emphasize that the primary contribution of our work is the introduction of a novel dataset that addresses a critical gap in standard object tracking, rather than proposing a new complex method for the task.
>
> To the best of our knowledge, TAO-Amodal is **the first large-scale amodal tracking benchmark** that features over 800 categories. TAO-Amodal also presents other contributions from its annotation protocols and comprehensive benchmarking analysis:
> - **Annotation Protocols.** Annotating amodal objects requires well-defined protocols tailored to various occlusion scenarios, such as in-frame and out-of-frame. The dynamic changes of object visibilities throughout the scene further complicate the annotation process. To address this, we identify possible visibility change scenarios and establish specific guidelines in Table 14. These structured annotation protocols not only standardize the development of future amodal tracking datasets but also identify various occlusion scenarios, providing valuable insights for advancing model development.
> - **Benchmarking analysis.** We benchmarked existing standard trackers and a line of amodal segmentation work. Additionally, we further include the evaluation results of open-vocabulary methods in this rebuttal. The comprehensive analysis highlights the challenging nature of amodal tracking and limitations of existing methods for handling occlusions. The empirical analysis also provides insights into which methods are suitable for developing amodal trackers on. For instance, with a global transformer architecture that cross-attends features from multiple frames, we noticed that GTR [1] performs better than other SOTA tracking methods.
>
> **Baselines in new tasks are often simple and effective.** Besides, we argue that *simplicity in methods is not inherently a drawback*, especially when the primary goal is to establish foundational baselines for a benchmark or a new domain like amodal tracking. Simple approaches often provide clarity and serve as a reference point for future research. An example could be drawn from 3D multi-object tracking (MOT), where Weng et al. [1] proposes a simple 3D MOT baseline combining a 3D Kalman filter and the Hungarian algorithm for data association. Despite the simplicity, this work has become an important baseline in 3D MOT. As shown in Table 3, amodal expander is more effective than other fine-tuning schemes, which verifies that amodal expander is a solid baseline in our TAO-Amodal benchmark.
>
> **Develop extensions of amodal expander.** Aside from the amodal expander, we also investigate its extensions by incorporating temporal signals using a Kalman filter and depth estimator. Additionally, we integrate multi-frame cross-attended Re-ID features into amodal expander. Our analysis suggests that we can further improve tracking metrics and detection metrics for heavily occluded objects, as detailed in Table 4. We hope this analysis could further address the reviewer’s concerns on our method design.
>
> *We believe that the value of our TAO-Amodal benchmark is not diminished by the simplicity of the method but is instead strengthened by the clarity and reproducibility it offers.* Thank you for your thoughtful discussion.
>
>
> [1] Weng et al. 3D Multi-Object Tracking: A Baseline and New Evaluation Metrics. IROS 2020.

---

### Official Review · Reviewer_RgK8 · 2024-11-03

**Soundness:** 3
**Presentation:** 3
**Contribution:** 3
**Rating:** 6
**Confidence:** 3

**Summary:**

The TAO-Amodal dataset method was introduced to address the amodal perception problem. It provides video sequences with 833 categories, including amodal and modal bounding boxes for visible and partially or fully occluded objects. This approach aims to address the scarcity of amodal benchmarks. Benchmarking current modal trackers and amodal segmentation methods reveals that existing methods struggle with detecting and tracking heavily occluded objects. Simple fine-tuning schemes were explored to mitigate this, improving the amodal tracking and detection metrics for occluded objects by 2.1% and 3.3%, respectively.

**Strengths:**

The authors present TAO-Amodal, a comprehensive and high-quality dataset designed to advance amodal perception by providing diverse occlusion annotations across 833 categories and introducing the Paste-and-Occlude (PNO) augmentation method, which simulates various occlusion scenarios to improve model robustness. This dataset enables rigorous benchmarking and reveals the limitations of current tracking algorithms under challenging conditions, offering a reliable foundation for future developments in amodal tracking research.

**Weaknesses:**

1、Limited Training Data: While the dataset is extensive for evaluation, the relatively small training set may restrict model generalization, especially for categories with fewer instances. This also means that subsequent research can only use this dataset for validation, making it ineffective for robust training.
2、During the data annotation phase, I noticed that the authors conducted detailed annotations, but I have a question: how did you determine the size of the occluded objects for annotation clearly?
3、The comparison is insufficient. The TAO dataset itself is an open-world dataset, but your evaluation method does not incorporate any open-world detection or open-world tracking approaches.
4、The examples of PasteNOcclude (PnO) shown in Figure 5 are overly simplistic. I want to know whether all data augmentation is done in this way.

**Questions:**

Please see the weakness. If my concerns are solved, I will consider raising my score.

---

> ### Author Response · Authors · 2024-11-18
> **Reply to reviewer Rgk8 (1/2)**
>
> Thank you for your review, we appreciate your comments that "TAO-Amodal offers a reliable foundation for future developments in amodal tracking research". We address your concerns below.
>
> &nbsp;
> > `Q1:` The evaluation should include open-world tracking methods.
>
> Thanks for the suggestion. We are happy to evaluate OVTrack [1], an open-vocabulary tracker that is capable of tracking any object classes by using vision-language models for classifications and associations. Please check our replies in the *Additional Evaluation Results* section below.
>
> &nbsp;
> > `Q2:` Limited Training Data
>
> We acknowledge that the training data in our dataset is relatively small. However, a high-quality evaluation benchmark could also significantly benefit the model training. We detail the reasons below.
>
> High-quality benchmarks drive the need for innovation in curating large-scale training data. This mirrors the evolution in large language models, where the introduction of challenging benchmarks [2, 3], has led to the collection of more comprehensive training data [4]. Some benchmarks [5] do not even have a training set and rely on the use of "internet as a training set” (c.f. Sec. 5).
>
> In computer vision, we are starting to see this paradigm shift with the use of synthetic data:
> - Amodal segmentation methods [6, 7, 8] create synthetic amodal data through pasting object segments onto images
> - Prominent modal tracking methods [9, 10] generate synthetic training videos through random cropping and resizing of static image datasets [11] to compensate for the lack of large vocabulary tracking data
> - State-of-the-art methods in 3D vision (e.g., monocular depth estimation [12] and scene flow [13]) use out-of-distribution synthetic training samples, and beat prior work on real-world evaluation
>
> Additionally, we note that despite the same small training set of the original TAO dataset, the modal tracking performance has increased from 10.2 to 27.5 TrackAP50 over the years [14], which includes performance increase on object categories where little to no training data existed.
>
> Finally, we provide some insights into the potential effects of scaling amodal training data through additional analysis in *Sec. B.2.1*. We provide further discussions on our dataset splits design and its benefits in *Sec. 3.2*.
>
>
> &nbsp;
> > `Q3:` How to determine the size of the occluded objects for annotation clearly?
>
> Our annotation protocol addresses the ambiguity in two ways:
> - We ask annotators to refer to both preceding and succeeding frames, adhering to the guidelines outlined in Table 14 for objects that are highly occluded. All occluded objects whose locations cannot be estimated from neighboring frames are marked with an `is_uncertain` flag, which account for < 1% of the data.
> - We also conduct a rigorous quality control process as detailed in Sec. 3. Specifically, each bounding box is annotated by one individual and undergoes two rounds of quality check by two separate annotators. Finally, a manual review by the authors of this work ensures the high quality of $> 99$% annotations, even for highly occluded objects. In other words, each bounding box is agreed upon by four different individuals.
>
> These efforts ensure that the annotations are as precise as possible, aligning with human-level accuracy, and ensure that uncertain annotations are correctly identified using the `is_uncertain` flag. We provide some examples for objects annotated with `is_uncertain` in Fig. 11.
>
>
> Despite the fact that it remains challenging to determine the real states of objects under heavy occlusions, we posit that annotations of human-level accuracy are sufficient to drive progress in amodal tracking. For example, when a basketball is placed under a box, humans can approximate the ball's overall position but find it difficult to determine its exact shape and state. This underscores both the inherent difficulty of amodal tracking and the idea that expecting models to produce more precise estimations than human annotators is likely too ambitious at this point. A more practical approach would be to focus on enabling models to approximate object states in a manner comparable to human perception as an initial step in advancing amodal tracking.

---

> ### Author Response · Authors · 2024-11-18
> **Reply to reviewer Rgk8 (2/2)**
>
> > `Q4:` Examples of PnO in Fig. 5 are overly simplistic.
>
> We provide more examples of PnO in Fig. 6 in our latest revised version. A characteristic of our PnO visualization is that object segments from the COCO and LVIS datasets are randomly selected and pasted onto the TAO-Amodal training video frames. This may alter the natural class distribution in individual videos. To address this, sampling strategies could be employed, such as identifying the object categories present in each training video and sampling object segments only from those categories. Additionally, this does not significantly affect the training of the amodal expander or the fine-tuning of the regression head. This is because the amodal expander is built on the pre-trained GTR [9] checkpoint, with the classification heads frozen during training. As a result, the model focuses solely on precise bounding box prediction. However, we do recognize the possibility that preserving the natural distribution of videos might give a slightly better performance.
>
>
>
> [1] Zhou et al. OVTrack. CVPR 2023. \
> [2] Zhang et al. MathVerse. arXiv 2024. \
> [3] Zhu et al. MathVista. ICLR 2024. \
> [4] Zhang et al. MAVIS. arXiv 2024. \
> [5] Hendrycks et al. MMLU. ICLR 2021. \
> [6] Li et al. Amodal instance segmentation. ECCV 2016. \
> [7] Zhu et al. Semantic amodal segmentation. CVPR 2017. \
> [8] Ozguroglu et al. pix2gestalt. CVPR 2024. \
> [9] Zhou et al. GTR. CVPR 2022. \
> [10] Zhou et al. CenterTrack. ECCV 2020. \
> [11] Gupta et al. LVIS. CVPR 2019. \
> [12] Ke et al. Marigold. CVPR 2024. \
> [13] Xiao et al. SpatialTracker. CVPR 2024. \
> [14] Dave et al. TAO Challenge, 2020.  https://motchallenge.net/data/TAO_Challenge/

---

> > ### Author Response · Authors · 2024-11-25
> > **Additional Evaluation Results**
> >
> > Dear Reviewer,
> >
> > We further evaluated the open-vocabulary tracker, OVTrack [1], on TAO-Amodal. We showed the results below:
> >
> >
> > | Method                  | AP$^{[0, 0.1]}$ | AP$^{[0.1, 0.8]}$ | AP$^{[0.8, 1.0]}$ | AP$^{OoF}$ | AP   | Track-AP | Track-AP$^{[0.8, 1.0]}$ |
> > |-------------------------|-----------|-------------|-----------|-------|------|----|-----------|
> > | OVTrack [1]    |  0.88  |  12.20   |   37.08    | 12.31| 26.60   | 15.00  | 6.99 |
> >
> >
> > &nbsp;
> > &nbsp;
> >
> > ---
> > **Analysis and insights.** OVTrack is an open-vocabulary tracker that is capable of tracking any object classes by using vision-language models for classifications and associations. Leveraging its ability to handle diverse object categories, OVTrack demonstrates strong results on visible objects. This highlights that open-vocabulary models are another interesting avenue for achieving open-world amodal tracking. OVTrack performs worse than GTR on tracking metrics for occluded objects, as shown in Table 2 of the paper, suggesting the significance of temporal cues. We hope these results could provide additional insights to inspire promising and intriguing future directions. We will update these numbers to Table 2 at the end of the discussion stage.
> >
> > &nbsp;
> > &nbsp;
> >
> > [1] Zhou et al. OVTrack. CVPR 2023.

---

> > > ### Comment · Reviewer_RgK8 · 2024-11-25
> > > **Reply**
> > >
> > > I'd like to thank the authors for all the effort to address my comments, questions, and concerns. I have no further questions and decide to raise my score.

---

> ### Author Response · Authors · 2024-11-25
> **Addressed Concerns and Open to Further Discussions**
>
> Dear Reviewer,
>
> We appreciate your valuable comments. We have made a significant effort to address your concerns, including the following:
> - Introduced additional evaluation results on the open-vocabulary methods to ensure a more comprehensive benchmarking.
> - Discussed the rationale behind our dataset splits design.
> - Illustrated how our dataset benefits future research in training data curation.
> - Discussed how we addressed the uncertainty in amodal tracking.
>
>
> We believe that these discussions have helped us improve the impact and clarity of our paper. We deeply appreciate your feedback and look forward to further discussions.
>
> We welcome any further feedback before the discussion deadline.
> Thank you!
>
> Best regards, Authors

---

### Official Review · Reviewer_MeLX · 2024-11-04

**Soundness:** 2
**Presentation:** 3
**Contribution:** 2
**Rating:** 5
**Confidence:** 3

**Summary:**

This work presents a large-scale video benchmarks for amodal tracking, and the evaluation metrics and baseline trackers are also designed.
The challenges of amodal detection and tracking by evaluating a number of amodal trackers and segmentors are validtated on the created benchmark.

**Strengths:**

1. A comprehensive annotation for amodal tracking based on TAO dataset is performed.

2. A plugin module for amodal tracking and some data augmentation methods are designed.

3. Extensive experiments and analysis are conducted.

**Weaknesses:**

1. The contributions of this work seem small. TAO dataset is existing, and the contribution to the benchmark is large-scale annotations. The designed expander is also a simple regressor and data augmentation schemes are also based on existing ones.

2. The motivation of this task is unclear to me. When an object is totally occluded, its state, including position, size and motion, is very difficult to predict. Although authors consume much time to annotate such objects, but the quality can not be guaranteed because we do not know their real states. What are the potential downstream applications or benefits of amodal tracking that motivate this work? How might uncertainty in amodal predictions be handled or utilized in subsequent tasks?

**Questions:**

The contributions of this work seem small and the motivation is unclear to me.

---

> ### Author Response · Authors · 2024-11-18
> **Reply to Reviewer MeLX (1/2)**
>
> Thank you for your review, we appreciate your comments that "annotations of TAO-Amodal are comprehensive and the experiments and analysis are extensive". We address your concerns below.
>
>
> &nbsp;
> >  Motivation of amodal tracking is unclear. What are the potential downstream applications or benefits of amodal tracking that motivate this work?
>
> Amodal tracking is especially crucial for tasks such as autonomous driving and robotics. For instance, detecting Vulnerable Road Users (VRUs), like pedestrians, benefits significantly from this capability. Tracking a pedestrian's movement even during temporary occlusions can greatly enhance safety and help prevent collisions. In contrast, standard tracking often fails in such occlusion scenarios because it is limited to focusing solely on visible parts of the object, which is shown in Figure 2 of the paper.
>
> Amodal tracking is also important in any applications that require the understanding of object permanence. For instance, in robotics, a robotic arm manipulator tasked with placing dishes into a dishwasher and retrieving them later may utilize amodal tracking to infer the plate’s existence. This capability ensures the robot can accurately track and interact with objects despite visibility challenges.
>
>
> &nbsp;
> > Contributions seem small and are limited to large-scale amodal annotations for the TAO dataset.
>
> Thank you for raising these concerns. Creating a large-scale amodal evaluation benchmark requires more than just annotation efforts. We clarify and point out other contributions for building TAO-Amodal below:
>
> **Design Annotation Protocols**
>
> * **Design annotation protocols:** Annotating amodal objects requires well-defined protocols tailored to the complexities of occlusion scenarios, such as in-frame and out-of-frame occlusions. The dynamic changes of object visibilities throughout the scene further complicate the annotation process. To address this, we identify possible visibility change scenarios and establish specific guidelines. For example, when an object is invisible before reappearing, we annotate those invisible frames only if this object was previously visible (from visible to invisible). Additionally, annotators are instructed to refer to both preceding and succeeding frames for objects that are highly occluded in the annotation guidelines. All occluded objects whose locations cannot be estimated from neighboring frames are marked with an `is_uncertain` flag, which account for < 1% of the data.
> * **Benefits of annotation protocols:** These structured annotation guidelines not only standardize the development of future amodal tracking datasets but also highlight challenging occlusion scenarios, providing valuable insights for advancing model development. We provide detailed annotation guidelines in Table 14 of our paper.
>
> **Design amodal evaluation metrics:**
>
> We design evaluation metrics tailored to different occlusion scenarios by utilizing estimated visibility attributes to classify occlusion levels into heavily occluded, partially occluded, out-of-frame, and non-occluded. This categorization helps identify the scenarios where models struggle to capture occluded objects. By adapting standard detection and tracking AP metrics, we further ensure the community can build amodal trackers on top of widely available standard trackers. We use 50\% IoU in our AP metrics as an attempt to curb the inherent uncertainty in the amodal detection and tracking tasks. This discussion can be found in Sec. 5.1 of the paper.
>
> **Benchmarking fine-tuned modal trackers and amodal baselines.**
>
> We benchmarked existing standard trackers by fine-tuning their regression heads on our training set and evaluating these methods on the validation set. We further benchmarked a line of amodal segmentation work. The comprehensive analysis suggests that even the SOTA trackers and amodal methods struggle to handle occlusions, highlighting the challenging nature of amodal tracking. The empirical analysis could also provide insights into which standard methods are suitable for developing amodal trackers. For instance, with a global transformer architecture that cross-attends features from multiple frames, we noticed that GTR [1] performs better than other SOTA tracking methods.

---

> ### Author Response · Authors · 2024-11-18
> **Reply to Reviewer MeLX (2/2)**
>
> > The method, amodal expander, is just a simple regressor, and PnO data augmentation is based on existing data augmentation schemes.
>
> The major goal in this work is to develop a high-quality evaluation benchmark while additionally providing insights into effective baselines (Table 3) and extensions, such as the integration of temporal information in Table 4, for amodal tracking. We outline the challenges and design choices for our methods below.
>
> We recognize that there is significant room for improvement in our methods, with the ultimate goal being the development of an end-to-end amodal tracker. However, *developing an end-to-end amodal tracker is highly challenging*, as evidenced by the performance degradation of fully fine-tuned standard trackers shown in Table 3.
>
>
> **Amodal expander and its extension.** To address this challenge, we introduced the amodal expander to bridge the gap by adapting the well-established community of standard trackers for amodal tracking. We built amodal expander on top of GTR, as this method shows strong results in both detection and tracking in Tab. 2, and observed performance gains over other fine-tuning schemes in Tab. 3. Using the expander as a plug-in module upon standard trackers might thus serve as an effective baseline for future work in developing stronger amodal tracking methods.
>
>
> To further support progress in amodal tracking, we conducted a comprehensive analysis exploring extensions of the amodal expander. For example, as shown in Table 4, we integrated temporal-aware features such as Kalman filter-based predictions, which improve detection of heavily occluded objects by utilizing prior position data. Additionally, incorporating ReID features through attention layers across multiple frames enhances tracking performance, demonstrating the potential of temporal cues.
>
>
> **PasteNOcclude (PnO).** Regarding PnO, we acknowledge its inspiration from augmentation techniques that paste objects onto images, widely used in detection and segmentation tasks. We adapted these techniques for amodal tracking to simulate occlusion scenarios, resulting in performance improvements, as shown in Table 4. This highlights that *properly adapting methods from the detection community is crucial for advancing amodal tracking*, especially given the current scarcity of training data in this domain.
>
>
> While there is room for refinement, we believe these approaches provide strong baselines and valuable insights to guide the advancement of amodal tracking methods. Finally, we provide a comprehensive discussion on the limitations and future directions in Sec. D of the appendix.
>
>
>
> &nbsp;
> > The quality of the annotations can not be guaranteed because we do not know their real states when an object is totally occluded. How might uncertainty in amodal predictions be handled or utilized in subsequent tasks?
>
> Our annotation protocol addresses the ambiguity in two ways:
> - We ask annotators to refer to both preceding and succeeding frames, adhering to the guidelines outlined in Table 14 for objects that are highly occluded. All occluded objects whose locations cannot be estimated from neighboring frames are marked with an `is_uncertain` flag, which account for < 1% of the data.
> - We also conduct a rigorous quality control process as detailed in Sec. 3. Specifically, each bounding box is annotated by one individual and undergoes two rounds of quality check by two separate annotators. Finally, a manual review by the authors of this work ensures the high quality of $> 99 $% annotations, even for highly occluded objects. In other words, each bounding box is agreed upon by four different individuals.
>
> We provide some examples for objects annotated with `is_uncertain` in Fig. 11.
>
> These efforts ensure that the annotations are as precise as possible, aligning with human-level accuracy, and ensure that uncertain annotations are correctly identified using the `is_uncertain` flag.  We posit that annotations of human-level accuracy are sufficient to drive progress in amodal tracking. For example, when a basketball is placed under a box, humans can approximate the ball's overall position but find it difficult to determine its exact shape and state. This underscores both the inherent difficulty of amodal tracking and the idea that expecting models to produce more precise estimations than human annotators is likely too ambitious at this point. A more practical approach would be to focus on enabling models to approximate object states in a manner comparable to human perception as an initial step in advancing amodal tracking.
>
>
> [1] Zhou et al. GTR. CVPR 2022.

---

> > ### Author Response · Authors · 2024-11-24
> > **Addressed Concerns and Open to Further Discussions**
> >
> > Dear Reviewer,
> >
> > We appreciate your valuable comments. We have made a significant effort to address your concerns, including the following:
> > - Elaborated the motivation behind amodal tracking and its downstream applications.
> > - Expanded the discussion on the annotation protocols of TAO-Amodal to highlight its contributions and quality.
> > - Provided additional insights and extensions of our methods.
> > - Discussed how we addressed the uncertainty in amodal tracking.
> >
> >
> > We believe that these discussions have helped us improve the clarity of our paper. We deeply appreciate your feedback and look forward to further discussions.
> >
> > We welcome any further feedback before the discussion deadline.
> > Thank you!
> >
> > Best regards,
> > Authors

---

> ### Author Response · Authors · 2024-11-26
> **Gentle Reminder**
>
> Dear Reviewer MeLX,
>
> We want to send you a gentle reminder that we have worked diligently to address your concerns. Please feel free to reach out if you have any further questions or require additional clarifications. We are more than happy to continue the discussion and strengthen our paper.
>
> Best regards,
> Authors

---

### Author Response · Authors · 2024-11-25
**Summary of Addressed Concerns and Open to Further Discussions.**

Dear Reviewers, ACs, and SACs,
We appreciate all the reviewers for their valuable comments and reviews. We have made a significant effort to address all the concerns, including the following:
- Discussed the motivation behind amodal tracking and its downstream applications.
- Expanded the discussion on the annotation protocols of TAO-Amodal to clarify its contributions and quality.
- Reiterated the rationale behind our dataset splits design.
- Provided additional insights and extensions of our methods to benefit future model development research.
- Outlined the primary obstacles of amodal tracking to present promising avenues for further exploration
- Introduced additional evaluation results on the open-vocabulary methods to ensure a more comprehensive benchmarking.


We believe that these discussions have helped us improve the clarity and impact of our paper. Once again, we deeply appreciate the feedback from all parties involved and look forward to further discussions.

We welcome any further feedback before the discussion deadline.
Thank you!

Best regards, Authors

---

### Note · Authors · 2025-01-22

I have read and agree with the venue's withdrawal policy on behalf of myself and my co-authors.